# Thinking in Flow: A Dissipative Stabilization Operator for Robust Autoregressive Reasoning

YuJie Huang [1]   WenWu He [1 2]   Zhuo-Xu Cui [3 4]

## Abstract

Chain-of-Thought (CoT) prompting enables multi-step reasoning in large language models, yet long-horizon generation remains brittle under distribution shift and context interference: irrelevant cues persist, small deviations compound into inference drift, and late-stage corrections can destabilize the trajectory. We recast autoregressive decoding as a perturbed long-horizon dynamical system and introduce a *dynamical stabilization operator* that targets *trajectory-level* reliability rather than token-level fluency. Specifically, we propose *ODE-guided language models*, which augment a base Transformer with a persistent continuous-time *thought state* whose dynamics are explicitly designed to be dissipative, enabling stable evidence accumulation with controlled forgetting. Instantiating this framework, *Thinking in Flow* (TiF) equips the model with a lightweight Neural ODE controller and injects its output through post-norm residual updates to achieve numerically stable, low-intrusion steering. A demand–supply (uncertainty–capacity) gate determines *when* intervention is warranted, while a direction gate determines *how* to steer in representation space, yielding selective, do-no-harm corrections instead of persistent bias. We establish well-posedness, dissipativity, and incremental stability of the controlled thought dynamics, implying bounded interventions over arbitrarily long contexts, and empirically demonstrate improved robustness to distractions and semantic perturbations, while matching

or improving accuracy on mathematical reasoning benchmarks across both the Llama and Qwen model families; we further observe gains on non-mathematical BBH reasoning tasks when training TiF on Llama. Code is available at https://github.com/MAiTL-Group/TIF.

## 1. Introduction

*"Consciousness, then, does not appear to itself chopped up in bits... it is nothing jointed; it flows."*
— William James, *The Principles of Psychology* (1890)

Chain-of-Thought (CoT) prompting has made it possible for large language models (LLMs) to solve problems that require multi-step reasoning (Wei et al., 2022). However, when generation unfolds over long horizons or under competing information, CoT becomes brittle: irrelevant context is difficult to ignore (Shi et al., 2023), small deviations accumulate into *inference drift*, and late-stage corrections are hard to apply without destabilizing the trajectory. Moreover, textual rationales are not always faithful to the model's internal decision process (Turpin et al., 2023; Dziri et al., 2023), motivating mechanisms that improve *trajectory-level* reliability rather than only producing fluent explanations.

**Autoregressive decoding as a stability problem.** A useful view is to treat decoding as a perturbed long-horizon dynamical system. At step $t$, a Transformer predicts the next token conditioned on the current hidden state and past tokens accessed through cached keys/values (the KV cache). Long-context studies reveal systematic utilization issues—e.g., positional attention bias (*lost-in-the-middle*) and "attention sinks"—that can cause relevant evidence to be underused while irrelevant cues persist (Liu et al., 2024; Hsieh et al., 2024; Xiao et al., 2024). However, the KV cache is an implicit memory rather than an explicit, task-adaptive stabilization law: it does not directly implement controlled forgetting or risk-triggered, calibrated corrections when the model is uncertain. Consequently, under distribution shift or context interference, small deviations can compound over long rollouts, making late-stage corrections difficult.

---

[1]School of Computing and Data Science, Fujian University of Technology, Fuzhou, Fujian, China [2]Fujian Provincial Key Laboratory of Big Data Mining and Applications, Fuzhou, Fujian, China [3]Shenzhen Institutes of Advanced Technology, Chinese Academy of Sciences, Shenzhen, Guangdong, China [4]Guangdong Provincial Key Laboratory of Multimodality Non-Invasive Brain-Computer Interfaces, Shenzhen, Guangdong, China. Correspondence to: WenwWu He <hwwhbb@163.com>, Zhuo-Xu Cui <zx.cui@siat.ac.cn>.

*Proceedings of the $43^{rd}$ International Conference on Machine Learning*, Seoul, South Korea. PMLR 306, 2026. Copyright 2026 by the author(s).

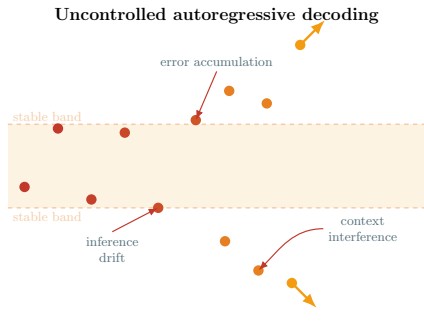

Uncontrolled autoregressive decoding

TiF: continuous thought-state stabilization

(a) Uncontrolled autoregressive decoding

(b) TiF: continuous thought-state stabilization

*Figure 1.* **Motivation: TiF as a stabilization operator for robust reasoning. (a) Uncontrolled drift:** Standard autoregressive decoding can accumulate small errors and context interference over discrete steps, yielding *inference drift* and departure from a desired reasoning trajectory (red). **(b) Continuous stabilization:** TiF models reasoning as a persistent *continuous thought-state* (blue) and applies *dissipative forgetting* together with *bounded intervention* to keep generation confined within a *stable band*—a bounded "tube" of solution-consistent trajectories over long horizons.

**Our proposal: a dynamical stabilization operator.** Figure 1 illustrates the motivation. In standard decoding, early deviations and context interference can push the generation trajectory outside a *stable band* (Figure 1(a)). We therefore introduce a lightweight *dynamical stabilization operator* that acts during decoding to keep the trajectory smooth and confined under distribution shift or context interference by combining controlled forgetting with bounded, selective corrections (Figure 1(b)). We learn its parameters by jointly fine-tuning the base Transformer and controller. Concretely, we propose *ODE-guided language models*, which augment a base Transformer with a persistent latent *thought state* and treat token-by-token decoding as observations of this state. The thought state evolves in continuous inner time and is *explicitly designed* to be dissipative, so that stale perturbations decay rather than persist, while interventions remain bounded and do-no-harm.

**Thinking in Flow (TiF): dissipative reasoning dynamics with selective intervention.** Instantiating this paradigm, TiF equips the Transformer with a lightweight Neural ODE controller that maintains a compact thought state and updates it by integrating the dissipative dynamics $\dot{s} = -\gamma s + g_\theta(s, \varphi_t)$ over an inner-time interval. The dissipative term implements controlled forgetting (contraction), while the driven term supports task-dependent evidence accumulation. The controller produces a steering signal that is injected through post-norm residual updates at each decoding step, enabling numerically stable, low-intrusion *selective corrections* that keep long-horizon decoding within the stable band depicted in Figure 1(b).

**Key design choices: risk-triggered, bounded steering.** A stabilization operator must intervene *only when needed* and *only to the necessary extent*. TiF achieves this via a demand–supply gate that couples a demand signal (un-

certainty) with a supply signal (controller-state capacity), yielding calibrated, do-no-harm interventions rather than persistent bias. Interventions remain bounded, preserving base behavior. To steer in a stable and interpretable way, TiF injects the controller signal in the model's readout space and modulates it with a lightweight direction gate, enabling sparse, direction-specific corrections. Formal definitions and implementation details are provided in Section 3.

**Why dissipative dynamics?** Dissipation provides a principled mechanism for long-horizon robustness: a contractive component attenuates the effect of past perturbations, mitigating inference drift as reasoning depth increases. We establish well-posedness, dissipativity, and incremental stability of the controlled thought dynamics, which implies bounded interventions and coherent steering over arbitrarily long contexts. We further provide an empirical contraction diagnostic for the implemented discrete flow map (Figure 3).

**Contributions.** We make three contributions:

- **Stabilization operator:** We recast autoregressive decoding as a perturbed long-horizon dynamical system and introduce a dynamical stabilization operator realized by ODE-guided language models with a persistent thought state, and train it with backpropagation.

- **Mechanism:** We propose dissipative reasoning dynamics with post-norm residual injection, combining uncertainty–capacity *impact gating* (when to intervene) with context-conditioned *direction gating* (how to intervene), yielding bounded, do-no-harm steering.

- **Theory & evidence:** We provide dynamical-systems guarantees (well-posedness, dissipativity, incremental stability), an empirical contraction diagnostic, and empirical gains in both accuracy and robustness under distractions and semantic perturbations.

## 2. Related Work

**Reasoning stabilization and sequence memory.** Chain-of-Thought prompting and related deliberative methods improve multi-step reasoning by exposing or searching over intermediate rationales (Wei et al., 2022; Kojima et al., 2022; Wang et al., 2023; Yao et al., 2023a;b), while recent work explores stronger supervision, latent reasoning, and implicit deliberation (Lightman et al., 2023; Cobbe et al., 2021; Hao et al., 2025; Zelikman et al., 2024). TiF is complementary to these approaches: rather than changing only the prompting format or training signal, it introduces a persistent controller state with an explicit dissipative evolution law to stabilize the decoding trajectory.

TiF is also related to long-context recurrence and state-space sequence models, including Transformer-style recurrence, compressed memory, SSMs, and gated linear/attention mechanisms (Dai et al., 2019; Rae et al., 2020; Sun et al., 2023; Gu & Dao, 2023; Gu et al., 2022; Poli et al., 2023). In particular, forget-gated linear models such as Gated DeltaNet use gates inside the sequence backbone to control recurrent memory mixing and enable targeted memory modification (Yang et al., 2025c). TiF shares the intuition that stale information should be attenuated, but differs in level and purpose: it wraps a pretrained Transformer with a lightweight auxiliary stabilization operator, using continuous-time dissipation to contract a compact thought state and an uncertainty–capacity gate to decide when to inject a bounded correction before the LM head. Thus, TiF does not replace the backbone memory mechanism; it regulates closed-loop reasoning trajectories. A fuller discussion of related reasoning, memory, and Neural ODE work is provided in Appendix A.

## 3. Methodology: TiF as a Dynamical Stabilization Operator

**Setup and objective: stabilizing autoregressive decoding.** Autoregressive decoding can be viewed as a perturbed long-horizon process: under irrelevant context, distribution shift, or small early mistakes, errors may compound and induce *trajectory drift*. We therefore introduce TiF as a *dynamical stabilization operator* that acts on the base model's per-step readout with *selective, bounded* interventions learned during joint fine-tuning. Concretely, TiF augments a base Transformer with a compact continuous-time *thought state* that (i) accumulates task-relevant evidence, (ii) attenuates stale signals via dissipative forgetting, and (iii) intervenes only when warranted through a gated residual injection in a scale-stable post-norm space. Figure 2 summarizes the overall TiF architecture.

**Notation.** At decoding step $t$, let $\mathbf{h}_t \in \mathbb{R}^{d_h}$ denote the normalized readout after the final LayerNorm (input to the LM head). TiF maintains a persistent controller state $\mathbf{s}_t \in \mathbb{R}^{d_s}$ with $d_s \ll d_h$. All bold symbols are vectors; $\odot$ denotes the Hadamard product.

**Operator view.** Let $\mathbf{z}_t = \mathbf{W}_{\text{head}}\mathbf{h}_t$ be the base logits. TiF produces a *stabilized* readout $\tilde{\mathbf{h}}_t$ (and logits $\tilde{\mathbf{z}}_t$) via

$$\tilde{\mathbf{h}}_t = \mathbf{h}_t + \underbrace{\alpha_t\big(\mathbf{r}_t \odot \mathbf{m}_t\big)}_{\text{risk-triggered, bounded steering}},$$

$$\tilde{\mathbf{z}}_t = \mathbf{W}_{\text{head}}\tilde{\mathbf{h}}_t.$$

Here $\alpha_t \geq 0$ is a demand–supply impact gate (Section 3.3) that makes the update nearly identity when the base model is confident, yielding *do-no-harm* behavior in low-risk regimes. The direction gate $\mathbf{r}_t$ is elementwise bounded (Section 3.2), ensuring direction-specific and scale-stable interventions in the post-norm space. The steering signal $\mathbf{m}_t$ is generated from a persistent thought state $\mathbf{s}_t$ whose inner-time dynamics are designed to be *dissipative* (Section 3.1), providing controlled forgetting that mitigates the accumulation of perturbations over long horizons. Together, these choices yield bounded, selective corrections that empirically reduce inference drift, rather than persistent bias.

### 3.1. From Discrete Tokens to a Persistent Continuous-Time Thought State

**Feature interface.** We compress the post-norm readout into a low-dimensional controller input:

$$\boldsymbol{\varphi}_t = \mathbf{U}_h\mathbf{h}_t, \tag{1}$$

where $\mathbf{U}_h$ is a learned projection. Crucially, $\boldsymbol{\varphi}_t$ is held *fixed* within the controller's inner-time evolution at step $t$, so that the controller implements a well-defined flow conditioned on the current token context.

**Dissipative Neural ODE (inner-time dynamics).** Given $(\mathbf{s}_t, \boldsymbol{\varphi}_t)$, TiF evolves the thought state over an inner-time interval $\tau \in [0, 1]$:

$$\frac{\mathrm{d}\mathbf{s}(\tau)}{\mathrm{d}\tau} = f_\theta(\mathbf{s}(\tau), \boldsymbol{\varphi}_t) = \underbrace{-\gamma\,\mathbf{s}(\tau)}_{\text{dissipative forgetting}} + \underbrace{g_\theta(\mathbf{s}(\tau), \boldsymbol{\varphi}_t)}_{\text{task-dependent drive}},$$

$$\mathbf{s}(0) = \mathbf{s}_t, \qquad \tau \in [0, 1], \tag{2}$$

where $\gamma > 0$ controls contraction and $g_\theta$ (a small MLP) injects context-dependent evidence. The dissipative term provides controlled forgetting, preventing unbounded state growth and mitigating drift over long horizons.

**Discrete flow map (implemented update).** The next-step controller state is the terminal value

$$\mathbf{s}_{t+1} = \mathbf{s}(1), \tag{3}$$

which we implement by numerical integration on $[0, 1]$. We denote the induced discrete flow map by

$$\mathbf{s}_{t+1} = \Psi(\mathbf{s}_t, \boldsymbol{\varphi}_t), \tag{4}$$

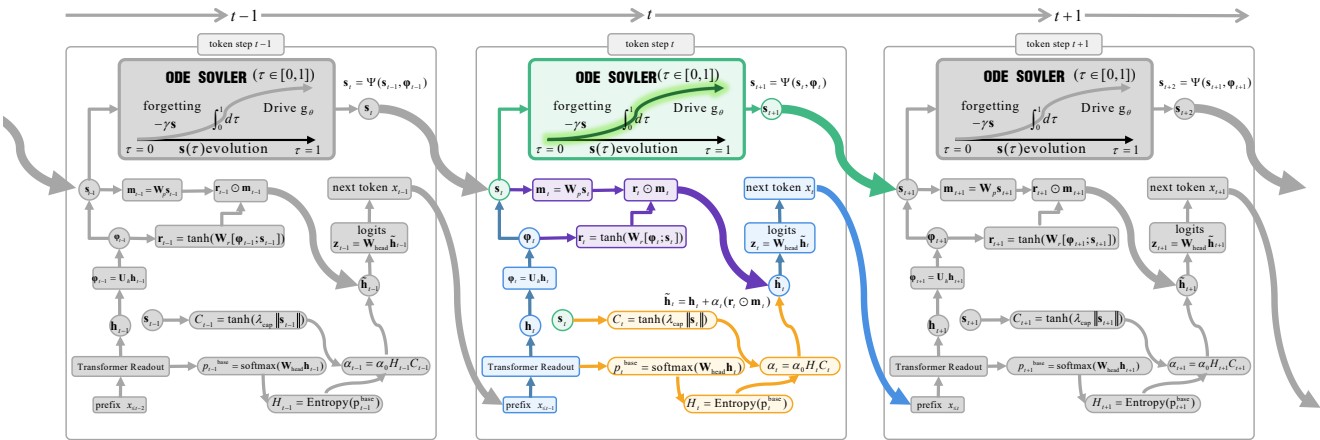

*Figure 2.* **TiF as a stabilization operator. (1) Context Interface (Blue):** The post-norm readout $\mathbf{h}_t$ encodes the current *context information* and is projected to a compact feature $\boldsymbol{\varphi}_t$. **(2) Continuous Dynamics (Green):** This feature drives a persistent thought state $\mathbf{s}_t$ via dissipative ODE dynamics ($\dot{\mathbf{s}} = -\gamma\mathbf{s} + g_\theta$), enforcing controlled forgetting to prevent drift. **(3) Risk-Triggered Gating (Yellow):** A demand–supply mechanism ($H_t, C_t$) determines *when* to intervene, ensuring corrections are applied only when the base model is uncertain. **(4) Bounded Injection (Purple):** The final steering vector is modulated by a direction gate $\mathbf{r}_t$ and injected back into the readout to stabilize the generation trajectory.

where $\Psi$ depends on the chosen solver (Euler/RK4/RK45). In all experiments we use a compute-lean fixed-step Euler solver for efficiency and fair comparisons, while higher-precision solvers are analyzed in ablations.

### 3.2. Post-Norm Residual Injection: How the Operator Steers

**Why post-norm injection.** We intervene in the *post-norm* readout space (after the final LayerNorm, before the LM head), which yields a normalized, scale-stable interface for consistent cross-token calibration. This reduces sensitivity to hidden-state scale variations and makes intervention magnitudes comparable across tokens and contexts.

**Direction-specific, bounded steering.** We map the controller state to the hidden space and apply an elementwise direction gate:

$$
\begin{aligned}
\mathbf{m}_t &= \mathbf{W}_p\,\mathbf{s}_t, \\
\mathbf{r}_t &= \tanh\left(\mathbf{W}_r[\boldsymbol{\varphi}_t;\mathbf{s}_t]\right), \\
\tilde{\mathbf{h}}_t &= \mathbf{h}_t + \alpha_t \cdot (\mathbf{r}_t \odot \mathbf{m}_t),
\end{aligned}
\tag{5}
$$

where $\mathbf{W}_p$ maps $\mathbf{s}_t$ to the LM hidden space and $\mathbf{r}_t$ selects steering directions with $\|\mathbf{r}_t\|_\infty \leq 1$. Thus, for any $t$, the intervention is *coordinate-wise bounded* by construction, and its overall magnitude is controlled by $\alpha_t$. Logits are then computed as $\mathbf{p}_t = \mathrm{softmax}(\mathbf{W}_{\mathrm{head}}\tilde{\mathbf{h}}_t)$.

### 3.3. Impact Gating: When and How Much to Intervene

A stabilization operator should be *risk-triggered* and *do-no-harm*: it should act only when the base model is uncertain, and remain nearly identity when the model is confident or the controller has not accumulated reliable signal. TiF

achieves this via a demand–supply coupling between an uncertainty-derived *demand* and a state-derived *supply*.

**Demand (uncertainty of the base prediction).** We compute uncertainty from the *pre-injection* distribution:

$$
\begin{aligned}
\mathbf{p}_t^{\mathrm{base}} &= \mathrm{softmax}(\mathbf{W}_{\mathrm{head}}\mathbf{h}_t), \\
H_t &= -\sum_{i=1}^{|\mathcal{V}|} p_{t,i}^{\mathrm{base}} \log p_{t,i}^{\mathrm{base}},
\end{aligned}
\tag{6}
$$

where $H_t$ is the entropy (with standard numerical stabilization).

**Supply (controller capacity / signal availability).** We measure whether the controller has accumulated meaningful signal via a bounded state-energy score:

$$
C_t = \tanh(\lambda_{\mathrm{cap}} \cdot \|\mathbf{s}_t\|_2),
\tag{7}
$$

where $\lambda_{\mathrm{cap}}$ is a learnable parameter, so that $C_t \approx 0$ when $\|\mathbf{s}_t\|_2$ is small (e.g., early in a rollout or when the controller remains inactive).

**Coupled impact gate (risk-triggered intervention magnitude).** The intervention magnitude is the multiplicative coupling

$$
\alpha_t = \alpha_0 \cdot H_t \cdot C_t,
\tag{8}
$$

with global scale $\alpha_0 \geq 0$. This product acts as a soft logical AND: intervention is substantial only when both (i) the base model is uncertain (high $H_t$) and (ii) the controller has non-negligible capacity (high $C_t$). Consequently, TiF is approximately identity in low-risk regimes, yielding do-no-harm behavior.

**Decoding procedure.** For completeness, the per-token TiF decoding procedure (ODE solve, uncertainty computation, and gated post-norm injection) is summarized in Algorithm 1 in Appendix D.2.

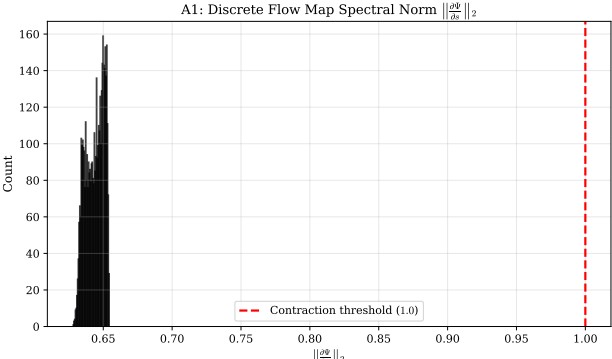

*Figure 3.* **Empirical contraction diagnostic of the implemented discrete-time update map.** We additionally compute a Jacobian spectral-norm diagnostic under a higher-precision RK4 implementation (fixed steps) as a conservative check on sampled inference-state distributions. The mass lies predominantly below 1.0, indicating local contraction of the discrete flow map on typical inference trajectories, without relying on global sufficient conditions.

# 4. Theoretical Analysis: Well-posed Stabilization and State-Space Stability

**Scope and what is stabilized.** We analyze TiF through the lens of a *stabilization operator* that acts during decoding by injecting a bounded, risk-triggered residual into the post-norm readout. Our theory focuses on the *controller state dynamics* that generate the steering signal: at each decoding step $t$, the feature $\varphi_t$ (Section 3) is treated as a bounded exogenous input held fixed over inner time $\tau \in [0, 1]$. Under standard regularity assumptions, we establish: (i) well-posedness of the inner-time ODE, (ii) dissipativity and an absorbing set for $\{s_t\}$, and (iii) incremental stability (input-to-state stability) that yields exponential forgetting of perturbations in $s_t$. Together with bounded gating, these results imply that TiF produces *bounded and coherent* steering signals over arbitrarily long contexts. We do *not* claim an unconditional global closed-loop stability theorem for token-level decoding. Instead, the main results establish state-space stabilization of the TiF controller, while Appendix E.8 provides a local closed-loop token-preservation guarantee: under local Lipschitz readout/steering assumptions and a positive stabilized-logit margin, bounded joint readout–state perturbations cannot flip the argmax token. Proofs are in Appendix E, and implementation-level contraction diagnostics for the discrete update map are reported in Figure 3 and Appendix B.3.

## 4.1. Assumptions
We treat the cognitive drive $g_\theta$ as an $L$-layer MLP in the state variable $s$, conditioned on $\varphi_t$.

**Assumption 4.1** (Regularity (uniform Lipschitz) and bounded offset). For all bounded $\varphi$ in a set $\Phi$, the map $s \mapsto g_\theta(s, \varphi)$ is globally Lipschitz with constant $L_s$ *uniform over* $\Phi$, i.e., $\|g_\theta(s_1, \varphi) - g_\theta(s_2, \varphi)\|_2 \leq L_s \|s_1 - s_2\|_2$ for all $s_1, s_2$ and all $\varphi \in \Phi$. Moreover, there exists $b_0 \geq 0$ such that $\|g_\theta(0, \varphi)\|_2 \leq b_0$ for all $\varphi \in \Phi$. Consequently, $\|g_\theta(s, \varphi)\|_2 \leq L_s \|s\|_2 + b_0$ for all $s$ and all $\varphi \in \Phi$.

**Assumption 4.2** (Bounded features and bounded gating). There exists $B_\varphi$ such that $\|\varphi_t\|_2 \leq B_\varphi$ for all $t$. With the impact gate definition (8) and the entropy/capacity bounds in Section 3.3, $\alpha_t$ is automatically bounded; we set $\alpha_{\max} := \alpha_0 \log |\mathcal{V}|$, hence $0 \leq \alpha_t \leq \alpha_{\max}$ for all $t$. In practice, bounded features can be encouraged by norm control on $U_h$ together with the normalized readout. We empirically verify feature-norm stability on sampled inference trajectories in Appendix B.3 (Figure 10).

Assumption 4.1 is standard in Neural ODE analyses and can be encouraged by norm control (e.g., spectral regularization or spectral normalization) (Miyato et al., 2018).

## 4.2. Existence and uniqueness on each decoding step
**Theorem 4.3** (Existence and uniqueness on the inner-time interval). *Fix a decoding step $t$ and context $\varphi_t$. Under Assumption 4.1, the initial value problem* (2) *admits a unique solution on $\tau \in [0, 1]$; hence the token-level update $s_{t+1} = s(1)$ in* (3) *is well-defined.*

Proof in Appendix E.2.

## 4.3. Dissipativity: an absorbing set for long-horizon decoding

**Theorem 4.4** (One-step dissipativity and a discrete absorbing set). *Assume Assumption 4.1 and let $\gamma > L_s$. Define $m_{\mathrm{diss}} := \gamma - L_s > 0$ and $\rho_{\mathrm{diss}} := e^{-m_{\mathrm{diss}}} \in (0, 1)$. Then for every decoding step,*

$$\|s_{t+1}\|_2 \leq \rho_{\mathrm{diss}} \|s_t\|_2 + (1 - \rho_{\mathrm{diss}}) \frac{b_0}{m_{\mathrm{diss}}}. \tag{9}$$

*Consequently, the state is uniformly bounded:*

$$\sup_{t \geq 0} \|s_t\|_2 \leq \max\left\{ \|s_0\|_2, \frac{b_0}{m_{\mathrm{diss}}} \right\}. \tag{10}$$

*Equivalently, for any $R > \frac{b_0}{m_{\mathrm{diss}}}$ the ball $\mathcal{B}_R$ is absorbing.*

Proof in Appendix E.3.

**Interpretation for stabilization.** Theorem 4.4 formalizes a key design requirement of a stabilization operator: the controller state that generates steering signals cannot grow without bound as the context length increases. Under $\gamma > L_s$, dissipativity enforces an absorbing set for $\{s_t\}$, providing controlled forgetting in the controller state and preventing long-horizon accumulation from inducing unbounded steering.

## 4.4. Trajectory coherence via incremental stability (ISS)

**Assumption 4.5** (Lipschitz dependence on context features). There exists $L_\varphi < \infty$ such that for all $\mathbf{s}$ and all bounded $\varphi, \varphi'$,

$$\|g_\theta(\mathbf{s}, \varphi) - g_\theta(\mathbf{s}, \varphi')\|_2 \le L_\varphi \|\varphi - \varphi'\|_2. \quad (11)$$

**Theorem 4.6** (Incremental stability and exponential forgetting). *Let $L_s$ be the Lipschitz constant from Assumption 4.1 and assume $\gamma > L_s$. Consider two token-level trajectories $\{\mathbf{s}_t\}$ and $\{\mathbf{s}'_t\}$ driven by feature sequences $\{\varphi_t\}$ and $\{\varphi'_t\}$, respectively. Define $m_{\text{ctr}} := \gamma - L_s > 0$ and $\rho_{\text{ctr}} := e^{-m_{\text{ctr}}} \in (0, 1)$. Then for every step $t$,*

$$\|\mathbf{s}_{t+1} - \mathbf{s}'_{t+1}\|_2 \le \rho_{\text{ctr}} \|\mathbf{s}_t - \mathbf{s}'_t\|_2 + \frac{1 - \rho_{\text{ctr}}}{m_{\text{ctr}}} L_\varphi \|\varphi_t - \varphi'_t\|_2. \quad (12)$$

Proof (and an unrolled bound) in Appendix E.4.

**Interpretation for stabilization.** Eq. (12) provides an input-to-state stability (ISS) bound for the controller: deviations in $\mathbf{s}_t$ decay geometrically at rate $\rho_{\text{ctr}} < 1$ up to a term proportional to the feature mismatch $\|\varphi_t - \varphi'_t\|_2$. This incremental stability is the mechanism-level counterpart of "trajectory coherence": small perturbations in the controller state are forgotten exponentially fast, which helps prevent long-horizon drift from persisting in the steering signal.

## 4.5. Implementation-level contraction diagnostic (discrete-time)

**Why we report an empirical diagnostic.** Theorems 4.4–4.6 rely on global sufficient conditions (e.g., $\gamma > L_s$) that may be conservative for the trained controller. To complement these guarantees, we probe the implemented discrete update map on states visited during inference. Appendix B.3 provides complementary diagnostics: it reports the distribution of the local Jacobian norm $\|\partial g_\theta / \partial s\|_2$ (as a proxy for $L_s$) relative to $\gamma$, and verifies bounded feature norms $\|\varphi_t\|_2$ on sampled inference trajectories in our representative diagnostic run. It also correlates $\|\varphi_t\|_2$ with the contraction coefficient of the implemented update to check robustness beyond low-norm inputs. Figure 3 shows that the Jacobian spectral norm is predominantly below 1, providing empirical evidence that the realized flow map is contractive in the region that matters (the reasoning manifold).

## 4.6. Bounded steering: from stable states to bounded interventions

**Corollary 4.7** (Bounded steering magnitude). *Under Assumptions 4.1 and 4.2, and the conditions of Theorem 4.4,*

$$\|\tilde{\mathbf{h}}_t - \mathbf{h}_t\|_2 = \|\alpha_t(\mathbf{r}_t \odot \mathbf{W}_p \mathbf{s}_t)\|_2$$
$$\le \alpha_{\max} \|\mathbf{W}_p\| \max\left\{ \|\mathbf{s}_0\|_2, \frac{b_0}{\gamma - L_s} \right\}. \quad (13)$$

**Implication for do-no-harm stabilization.** Corollary 4.7 links controller-state stability to the magnitude of the readout intervention. Since $\alpha_t$ is risk-triggered and bounded (Assumption 4.2) and $\{\mathbf{s}_t\}$ remains in an absorbing set (Theorem 4.4), TiF cannot inject arbitrarily large perturbations into the base model. This provides a mechanistic explanation for why TiF behaves as a stabilization operator: the intervention is both *selective* (through gating) and *bounded* (through dissipativity and norm control).

**Extensions.** Appendix E provides (i) an unrolled incremental-stability bound (Appendix E.4), (ii) a steering-stability bound that accounts for the direction gate (Appendix E.5), and (iii) a discussion of how norm control helps satisfy the Lipschitz conditions (Appendix E.7).

**From bounded steering to token-level preservation.** The above results show that the TiF controller produces bounded and stable steering signals. To address the remaining gap between state-space stability and token-level reasoning, Appendix E.8 further provides a local closed-loop preservation result. In a neighborhood of a reference correct reasoning trajectory, under local Lipschitz readout/steering assumptions and a positive stabilized-logit margin, the joint readout–controller error remains bounded by a Schur-stable comparison system. Consequently, if the induced logit perturbation is smaller than half of the reference margin, the argmax token cannot flip at any decoding step. This yields a conditional token-level guarantee: TiF preserves a correct reasoning chain under bounded local perturbations, rather than claiming unconditional global accuracy improvement.

# 5. Experiments
## 5.1. Experimental Setup

**Setup.** We evaluate on **GSM8K** (Cobbe et al., 2021) (7K train, full test) and **MATH** (Hendrycks et al., 2021) (7K train, evaluated on **MATH500** (Lightman et al., 2023)) using four open-weight models: **Llama-3.2-3B**, **Qwen3-4B** (Yang et al., 2025a), **Qwen2.5-3B** (Yang et al., 2024) , and **Llama-3.1-8B** (Dubey et al., 2024). We report accuracy with **Greedy Decoding (greedy@1)** and **Self-Consistency Majority Voting (maj@8)**. We adopt a two-stage training strategy: first freeze the base LLM and train only the ODE controller, then jointly fine-tune both components. We use a fixed-step Euler solver ($N = 1$) by default. For fair comparison *within each base model*, all methods (SFT, TiF, and other baselines) use the same training configuration and the same training budget, and are evaluated at the same reporting checkpoint (the same epoch) for that base model. We use a fixed budget of 10 epochs for both datasets; the resulting number of optimizer updates differs across datasets due to different effective batch sizes (see Appendix D.3).

**Baselines.** We compare TiF against: (1) **SFT**: Standard supervised fine-tuning; (2) **MFT** (Chen et al., 2024): Structure-

*Table 1.* Main results for pretrained *base* LMs (Qwen3-4B, Qwen2.5-3B, Llama-3.2-3B, and Llama-3.1-8B). We report **in-domain** performance: models fine-tuned on GSM8K are evaluated on GSM8K (test), and models fine-tuned on MATH are evaluated on MATH500. We report accuracy (%) with greedy decoding (greedy@1) and self-consistency majority vote over 8 samples (maj@8), along with their **95% Confidence Intervals (CI)**(Computed from the test set using the normal approximation to a binomial proportion). For TiF, we append the absolute accuracy change vs. the corresponding SFT baseline in parentheses (green: gain; red: drop).

| Base model | Method | GSM8K (test) | | MATH500 | |
| --- | --- | --- | --- | --- | --- |
| | | greedy@1 | maj@8 | greedy@1 | maj@8 |
| Qwen3-4B | SFT | $77.41_{[75.15,79.66]}$ | $86.58_{[84.74,88.42]}$ | $47.20_{[42.82,51.58]}$ | $56.00_{[51.65,60.35]}$ |
| | TiF (ours) | $77.94_{[75.70,80.18]}$ (+0.53) | $88.02_{[86.27,89.77]}$ (+1.44) | $47.60_{[43.22,51.98]}$ (+0.40) | $58.60_{[54.28,62.92]}$ (+2.60) |
| Qwen2.5-3B | SFT | $63.68_{[61.09,66.28]}$ | $76.42_{[74.13,78.71]}$ | $33.60_{[29.46,37.74]}$ | $44.00_{[39.65,48.35]}$ |
| | TiF (ours) | $66.34_{[63.79,68.89]}$ (+2.66) | $76.72_{[74.44,79.01]}$ (+0.30) | $36.80_{[32.57,41.03]}$ (+3.20) | $46.40_{[42.03,50.77]}$ (+2.40) |
| Llama-3.2-3B | SFT | $44.73_{[42.05,47.41]}$ | $53.30_{[50.61,55.99]}$ | $9.00_{[6.49,11.51]}$ | $14.20_{[11.14,17.29]}$ |
| | TiF (ours) | $50.49_{[47.79,53.19]}$ (+5.76) | $58.38_{[55.72,61.04]}$ (+5.08) | $11.00_{[8.26,13.72]}$ (+2.00) | $15.20_{[12.05,18.35]}$ (+0.60) |
| Llama-3.1-8B | SFT | $63.84_{[61.24,66.43]}$ | $68.01_{[65.49,70.52]}$ | $19.20_{[15.75,22.65]}$ | $22.60_{[18.93,26.27]}$ |
| | TiF (ours) | $64.29_{[61.71,66.88]}$ (+0.45) | $68.54_{[66.03,71.04]}$ (+0.43) | $19.40_{[15.93,22.87]}$ (+0.20) | $22.60_{[18.93,26.27]}$ (0.00) |

*Table 2.* Comparison with baselines on GSM8K (test) and MATH500. All methods use **Llama-3.2-3B** as the base model. Models are fine-tuned on either GSM8K (evaluated on GSM8K) or MATH (evaluated on MATH500). We report greedy decoding (greedy@1) and self-consistency majority vote over 8 samples (maj@8). **Bold** indicates the best performance.

| Method | GSM8K (test) | | MATH500 | |
| --- | --- | --- | --- | --- |
| | greedy@1 | maj@8 | greedy@1 | maj@8 |
| SFT | 44.73 | 53.30 | 9.00 | 14.20 |
| NEFTune | 45.49 | 53.70 | 8.20 | 13.60 |
| Pause (Prefix) | 42.84 | 52.99 | 9.00 | 13.40 |
| Pause (Dense) | 43.06 | 53.36 | 8.40 | 11.20 |
| MFT | 49.58 | 57.77 | 7.20 | 10.80 |
| RE-Control | 40.18 | 47.99 | 10.00 | 12.40 |
| TiF (ours) | **50.49** | **58.38** | **11.00** | **15.20** |

*Table 3.* Accuracy (%) on irrelevant-context robustness benchmarks: GSM-IC-4K (Shi et al., 2023) and GSM-DC-with-IC (Yang et al., 2025b). Base model: Llama-3.2-3B fine-tuned on GSM8K. Scores are reported with greedy decoding. For TiF, we append the absolute accuracy change vs. the SFT baseline in parentheses (green: gain; red: drop).

| Method | GSM-IC-4K | GSM-DC-with-IC | | |
| --- | --- | --- | --- | --- |
| | Acc. | Light | Medium | Hard |
| SFT | 53.83 | 6.05 | 5.29 | 2.90 |
| **TiF** | 55.25 (+1.42) | 6.86 (+0.81) | 5.38 (+0.09) | 3.67 (+0.77) |

aware masking; (3) **NEFTune** (Jain et al., 2024): Regularization via embedding noise; (4) **Pause Token** (Goyal et al., 2024): Computation injection via **Prefix** (3 tokens) or **Dense** (token-wise) insertion; and (5) **RE-Control** (Kong et al., 2024): a representative test-time representation control baseline.For RE-Control, we follow the algorithm and default

hyperparameters described in Kong et al. (2024), using our GSM8K-SFT checkpoint as the starting point.

### 5.2. Main Results: Accuracy and Consistency

Table 1 demonstrates that TiF improves or maintains performance across the evaluated model families (Llama and Qwen) and scales (3B–8B), suggesting good transferability across architectures. However, the gains are scale- and difficulty-dependent: they tend to be larger on smaller models and on harder, longer-horizon instances, while average improvements on larger models can be modest. Consistent with incremental stability (Theorem 4.6), Figure 4 shows that benefits concentrate on the longest reasoning chains, where inference drift is most severe. For the larger Llama-3.1-8B model, depth-stratified GSM8K results and paired flip analysis are provided in Appendix B.2.

**Robustness in Long Reasoning Chains.** To analyze whether TiF effectively mitigates inference drift, we stratified the GSM8K test set by reasoning depth (Figure 4). As expected, absolute accuracy for both models decreases as problem complexity increases (Left). However, the *relative benefit* of the ODE controller becomes increasingly significant in longer trajectories (Right). While the improvement is +6.9% on short tasks (2 steps), it surges to **+25.3%** on the longest tasks ($\geq 5$ steps). This trend suggests that standard SFT is disproportionately brittle in long-horizon settings due to error accumulation, whereas TiF's dissipative dynamics provide the necessary stability to sustain coherent reasoning in the most demanding scenarios.

### 5.3. Comparison with Baselines

As shown in Table 2, TiF outperforms baselines across three distinct paradigms. **Pause Tokens** (computation injection) fail to improve over SFT, indicating that discrete token padding cannot effectively extend reasoning depth

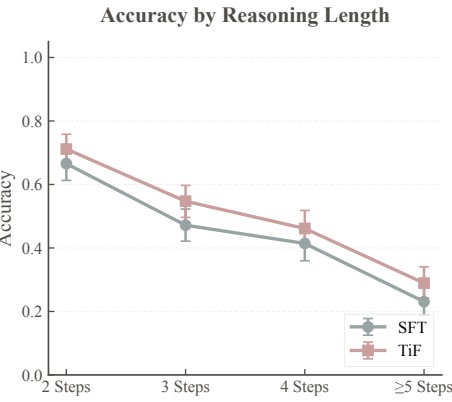
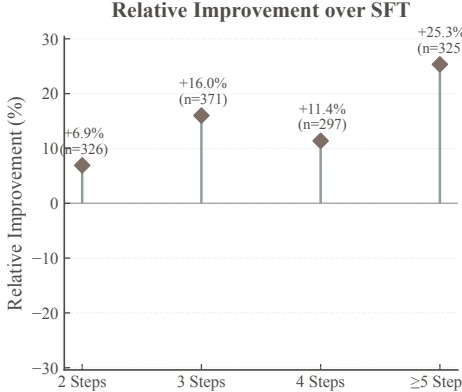

*Figure 4.* **Resilience in Long-Horizon Reasoning. (Left)** Absolute accuracy on GSM8K stratified by reasoning depth. While both models face increased difficulty as steps accumulate, TiF maintains a consistent performance lead across all depths. **(Right)** The relative improvement over SFT becomes most pronounced in the longest reasoning chains, peaking at **+25.3%** for problems requiring $\geq 5$ steps. This indicates that TiF is particularly effective at mitigating failure in high-complexity scenarios where inference drift is most severe.

without a persistent memory state. **NEFTune** (regularization) offers marginal gains (+0.76%) via embedding noise, but lacks structural guidance. The strongest competitor is **Masked Thought (MFT)**, which implicitly encourages robustness via partial masking. However, TiF surpasses MFT by explicitly modeling the reasoning process as a stable dynamical system, demonstrating that *guided continuous evolution* yields better trajectory consistency than random noise (NEFTune) or implicit masking (MFT).

### 5.4. Robustness Analysis

We evaluate robustness under (i) irrelevant-context distractors and (ii) semantic perturbations. TiF improves over SFT on GSM-IC and GSM-DC-with-IC (Shi et al., 2023; Yang et al., 2025b), with the largest gain on the **Hard** split of GSM-DC-with-IC (+0.77%) (Table 3). On GSM-Plus (Li et al., 2024), TiF attains the best accuracy in 6/7 categories and remains competitive on the remaining one; the largest gains are on *Numerical Substitution* (+5.00%) and *IDF Conversion* (+4.86%) (Figure 5). These results align with the stability motivation (Theorem 4.4, Theorem 4.6) and our empirical contraction diagnostic (Figure 3).

### 5.5. General Reasoning: Non-Math Tasks

In addition to the mathematical reasoning tasks (GSM8K, MATH), we evaluate TiF on general symbolic and algorithmic reasoning using a curated subset of 6 tasks from the BIG-Bench Hard (BBH) benchmark (Suzgun et al., 2023). Specifically, we include *Logical Deduction* (3, 5, and 7 objects), *Reasoning about Colored Objects*, and *Tracking Shuffled Objects* (3 and 5 objects). We selected these tasks to isolate the model's core inference capabilities from knowledge retrieval or linguistic nuances. They serve as a rigorous stress test for the model's deductive engine, specifically quantifying its capacity for multi-hop logical transitions,

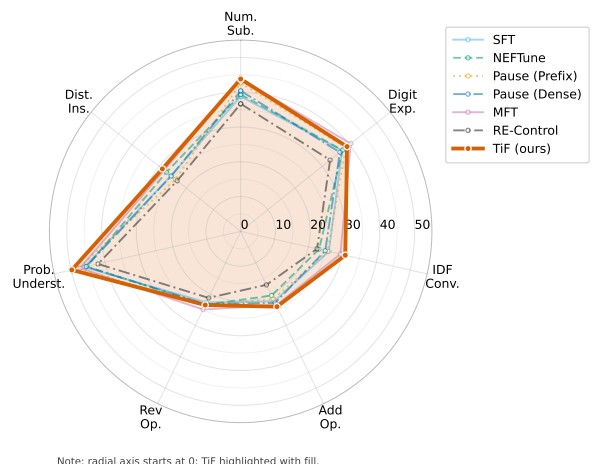

*Figure 5.* **Adversarial Robustness on GSM-Plus.** Radar chart comparison of TiF vs. baselines across 7 perturbation categories. TiF (blue polygon) consistently covers a larger area than baselines (SFT, Pause, MFT), indicating superior robustness across diverse semantic perturbations. Detailed numerical results are provided in Section C.2.

complex constraint satisfaction, and dynamic state tracking within the working memory. For detailed descriptions of these tasks, please refer to Appendix C.3.

**Training Setup.** For these non-mathematical tasks, we employ a distilled dataset to train the TiF controller on **Llama-3.2-Base**. Specifically, we use the CSQA Level 6 ChatGPT dataset from Chen et al. (2025), which contains synthetic Chain-of-Thought rationales generated with high granularity (Level 6). This dataset allows us to verify the generalization of our latent space control mechanism to broader reasoning domains beyond mathematics. We use a

distilled CoT dataset primarily for *format alignment*: it provides step-structured traces and a consistent multiple-choice answer interface (e.g., A/B/C/D), which matches BBH-style evaluation and enables reliable automatic parsing.

**Results.** Table 4 reports accuracy on the 6-task BBH subset used in our evaluation. TiF surpasses SFT across all six logical reasoning tasks.

*Table 4.* Performance comparison on the 6-task BBH subset. We report accuracy (%) for SFT baselines and TiF. Best results are bolded.

| Task | SFT | TiF |
|------|-----|-----|
| Logical Deduction (3 Objects) | 37.6 | **42.0** |
| Logical Deduction (5 Objects) | 29.6 | **31.2** |
| Logical Deduction (7 Objects) | 26.8 | **29.6** |
| Reasoning about Colored Objects | 38.4 | **39.6** |
| Tracking Shuffled Objects (3 Objects) | 30.8 | **30.8** |
| Tracking Shuffled Objects (5 Objects) | 16.8 | **17.6** |

## 5.6. Ablation Study

We ablate key components of TiF on GSM8K (Table 5). To isolate the benefit of persistent dynamics from the mere effect of adding tunable parameters, we include a parameter-matched non-recurrent residual baseline. This baseline adds a lightweight residual projection before the LM head with the same additional parameter budget as TiF (+2.08M), but does not maintain a recurrent controller state across decoding steps. It improves over SFT, showing that extra capacity is helpful, but remains below TiF by 2.12 points in greedy@1 and 0.84 points in maj@8. This gap indicates that TiF's gains are not solely due to increased capacity; the persistent dissipative state dynamics provide additional benefit.

Removing the entropy-based demand $H_t$ or capacity $C_t$ from the impact gate (Eq. 8) degrades performance, confirming that uncertainty-triggered, supply-modulated intervention is critical. Removing the direction gate $\mathbf{r}_t$ (Eq. 5) also hurts accuracy, indicating that elementwise steering enables more precise corrections. Replacing the ODE state update with a comparably-sized GRU (Chung et al., 2014) (all else fixed) yields the largest drop among recurrent-update variants.

## 6. Conclusion

We proposed *Thinking in Flow* (TiF), a dynamical stabilization operator for autoregressive reasoning that augments a Transformer with a lightweight dissipative controller maintaining a persistent thought state. By making controlled forgetting and risk-triggered, bounded intervention explicit, TiF mitigates inference drift and improves robustness in long-horizon generation. Our analysis establishes well-posedness, dissipativity, and incremental stability of the

*Table 5.* Ablation study on GSM8K with Llama-3.2-3B. The non-recurrent residual baseline uses the same additional parameter budget as TiF (+2.08M) and injects a lightweight residual projection before the LM head, but does not maintain or pass a recurrent controller state across decoding steps.

| Method | greedy@1 | maj@8 |
|--------|----------|-------|
| SFT Baseline | 44.73 | 53.30 |
| Non-Recurrent Residual | 48.37(+3.64) | 57.54(+4.24) |
| TiF (Full) | **50.49**(+5.76) | **58.38**(+5.08) |
| w/o $H_t$ | 46.85(-3.64) | 56.71(-1.67) |
| w/o $C_t$ | 47.84(-2.65) | 55.65(-2.73) |
| w/o $\mathbf{r}_t$ | 47.92(-2.57) | 56.56(-1.82) |
| ODE $\rightarrow$ GRU | 45.67(-4.82) | 54.21(-4.17) |

controller dynamics under sufficient conditions, and our empirical contraction diagnostic is consistent with locally contractive behavior on sampled inference trajectories. Across multiple open-weight LLMs and reasoning benchmarks, TiF yields gains especially on harder, longer-horizon instances and under distractors or semantic perturbations. Future work includes improving the efficiency of the controller update (e.g., more efficient or closed-form state evolution) and using better-calibrated risk signals beyond entropy to intervene on confidently wrong predictions.

## Acknowledgements

This work was supported by the Shenzhen Science and Technology Program (Grant No. JCYJ20240813155840052), the National Natural Science Foundation of China (Grant Nos. 62476268 and 62206273), the Guangdong Provincial Key Laboratory of Multimodality Non-Invasive Brain-Computer Interfaces (Grant No. 2024B1212010010), and the Natural Science Foundation of Fujian Province of China (Grant No. 2024J01158).

## Impact Statement

This paper presents work whose goal is to advance the field of machine learning. There are many potential societal consequences of our work, none of which we feel must be specifically highlighted here.

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

# Appendix Table of Contents

# A. Additional Related Work

**Chain-of-Thought prompting and deliberative inference.** Chain-of-Thought (CoT) prompting exposes intermediate reasoning steps and substantially improves multi-step problem solving in LLMs (Wei et al., 2022; Kojima et al., 2022; Wang et al., 2023). Beyond standard CoT, prompting and decoding schemes have been proposed to better allocate computation or decompose difficulty, including least-to-most prompting (Zhou et al., 2023), external scratchpads for intermediate computation (Nye et al., 2021), and self-training style bootstrapping such as STaR (Zelikman et al., 2022). More recent search-style inference frameworks (e.g., Tree-of-Thoughts) explicitly explore a space of partial rationales (Yao et al., 2023a), while ReAct interleaves reasoning with action/observation for improved control (Yao et al., 2023b). However, long-horizon rationales can remain brittle and may be unfaithful to the model's internal decision process (Turpin et al., 2023; Dziri et al., 2023), motivating trajectory-level stabilization rather than purely richer textual rationales.

**Training-time interventions for reasoning quality and robustness.** A complementary line of work improves reasoning through fine-tuning objectives or regularization. For mathematical reasoning, process-level supervision and verifiers (or stepwise verification) provide stronger learning signals than pure outcome supervision (Lightman et al., 2023; Cobbe et al., 2021). Recent training recipes target robustness and generalization, including masking-based reasoning objectives such as Masked Thought (Chen et al., 2024), embedding-noise regularization like NEFTune (Jain et al., 2024), and computation injection via pause tokens (Goyal et al., 2024). Our approach differs by introducing an explicit *persistent* thought state with

a designed evolution law; rather than changing only the supervision signal, we impose dynamical structure (dissipation and contraction) that directly targets long-horizon drift.

**Latent, implicit, and continuous-time reasoning states.** Several recent works argue that effective reasoning may occur in *latent* space rather than being fully verbalized. Training LLMs to reason in a continuous latent space (Hao et al., 2025) and "quiet" internal reasoning traces (Zelikman et al., 2024) both emphasize separating internal computation from surface text. In parallel, implicit-depth models such as Deep Equilibrium Models (DEQ) compute fixed points of iterative transformations, effectively enabling infinite-depth computation with constant-memory backpropagation (Bai et al., 2019). TiF is aligned with this theme in that it treats reasoning as the evolution of a *state*, but differs by (i) using an explicit dissipative Neural ODE controller (Chen et al., 2018) for *continuous-time* state evolution and (ii) coupling the controller to the base LM through selective, uncertainty-aware residual injection.

**Long-context memory, recurrence, and state-space sequence models.** Architectures for long contexts often focus on improving *memory access* or *efficient recurrence*. Transformer-XL introduces segment-level recurrence to extend context beyond a fixed window (Dai et al., 2019), and the Compressive Transformer further augments this with a compressed memory for longer horizons (Rae et al., 2020). Recurrent Memory Transformers (Bulatov et al., 2022) and streaming tricks such as attention sinks (Xiao et al., 2024) pursue related goals. More broadly, state-space and alternative sequence models (e.g., RetNet, Mamba) replace or modify attention to improve long-range efficiency (Sun et al., 2023; Gu & Dao, 2023), with structured state-space models like S4 providing a principled long-sequence backbone (Gu et al., 2022) and long-convolution operators such as Hyena scaling to very long contexts with gating (Poli et al., 2023). Recent forget-gated linear models make this distinction sharper. For example, Gated DeltaNet combines gating for adaptive memory control with delta-rule updates for targeted memory modification (Yang et al., 2025c). This is conceptually related to TiF in that both recognize the need to attenuate stale or interfering information. The key difference is architectural and functional: Gated DeltaNet uses native discrete-time gates inside the sequence backbone to manage recurrent memory mixing, whereas TiF introduces a lightweight auxiliary controller on top of a Transformer and uses continuous-time dissipation to stabilize a compact thought state. Moreover, TiF's impact gate does not primarily decide which backbone memory to retain; it couples base-model uncertainty with controller capacity to decide when and how strongly to inject a bounded correction into the decoding trajectory. Thus, TiF targets trajectory-level stabilization of autoregressive reasoning rather than general-purpose efficient sequence modeling.

**Neural ODEs, stable networks, and contraction principles.** Viewing deep networks as dynamical systems has a long tradition; Neural ODEs formalize continuous-depth computation (Chen et al., 2018), and follow-ups such as Augmented Neural ODEs improve expressivity (Dupont et al., 2019). Stability-motivated architectures treat depth as a time discretization and impose energy/stability structure (Haber & Ruthotto, 2017), while controlled differential equation formulations provide a general continuous-time modeling toolkit for irregular inputs (Kidger et al., 2020). From a control-theoretic perspective, contraction analysis characterizes incremental stability via uniform shrinking of perturbations (Lohmiller & Slotine, 1998), complementing classic nonlinear systems treatments (Khalil, 2002; Coddington et al., 1956). TiF builds directly on these ideas: we design the controller dynamics to be dissipative and incrementally stable, yielding bounded, coherent steering signals over long decoding horizons.

**Robustness to irrelevant context and long-horizon degradation.** LLMs can be significantly distracted by irrelevant context (Shi et al., 2023; Yang et al., 2025b), and robustness benchmarks (e.g., GSM-Plus) reveal systematic failure modes under controlled perturbations (Li et al., 2024). Long-context usage itself can degrade in nontrivial ways; "lost-in-the-middle" effects show retrieval failures when key information is placed mid-context (Liu et al., 2024). These findings motivate methods that do more than enlarge context windows: they require mechanisms that stabilize the *trajectory* of inference under distraction and perturbation. TiF addresses this by combining selective intervention (uncertainty–capacity gating) with dissipative state evolution that attenuates early perturbations.

## B. Analysis, Diagnostics, and Mechanism

### B.1. Mechanism Analysis: Dynamics of the Dissipative Controller

Figure 6 visualizes how the DOC operates during generation. We address two questions: *when* does it intervene (macro-level $\alpha_t$), and *how* does it steer (micro-level injections).

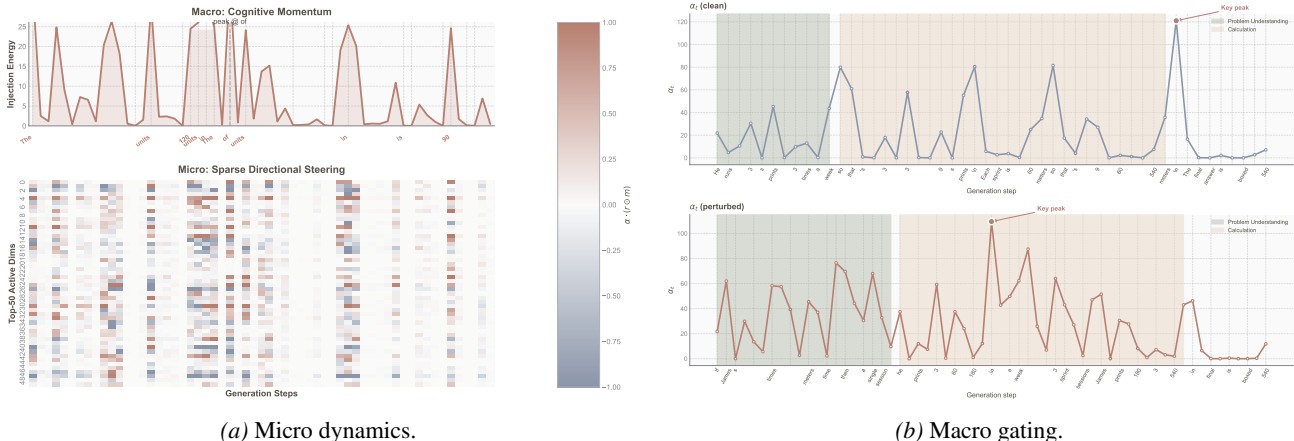

*(a)* Micro dynamics.                                                                                    *(b)* Macro gating.

*Figure 6.* Mechanism analysis of the Dissipative ODE Controller (DOC), contrasting micro-level steering with macro-level gating. **(Left)** The injected term $\alpha_t(\mathbf{r}_t \odot \mathbf{m}_t)$ from Eq. (5) exhibits pulse-like spikes at information-dense tokens and sparse, signed activation across dimensions, indicating direction-specific rather than uniform steering. **(Right)** The impact gate $\alpha_t$ from Eq. (8) is compared between a clean math problem (top) and its perturbed variant embedded in an elaborate narrative context with unchanged calculation logic (bottom); the controller shows early suppression of distractors and structure-aware peaks at step boundaries, with later-stage dynamics aligning across both runs.

**When does it intervene?**    Figure 6b compares $\alpha_t$ on a standard math problem (top) versus the same problem embedded in an elaborate story context that preserves the underlying calculation (bottom). The impact gate $\alpha_t$ peaks at logical connectors and step boundaries (e.g., newline tokens), suggesting structure-aware gating. Under the perturbed prompt, $\alpha_t$ exhibits a pronounced early peak while processing the narrative distractor, followed by realignment with the clean trajectory in later reasoning steps. This behavior is consistent with the incremental stability in Theorem 4.6: early perturbations are dissipated, and the controller locks onto reasoning structure rather than surface noise.

**How does it steer?**    Figure 6a shows that injection energy concentrates into brief, pulse-like events at information-dense tokens rather than remaining constantly active. The injected vector $\alpha_t(\mathbf{r}_t \odot \mathbf{m}_t)$ is sparse across dimensions with signed (positive/negative) activations, indicating selective enhancement and suppression of specific coordinates. This sparse, direction-specific steering via Eq. (5) enables low-intrusion corrections without uniform amplification.

**Limitations.**    TiF is explicitly designed as a *risk-triggered* and *do-no-harm* stabilization operator: the intervention magnitude is gated by uncertainty and controller capacity, $\alpha_t = \alpha_0 H_t C_t$. As a consequence, when the base model is already confident (low entropy), TiF naturally applies near-identity updates and the average gains can be small. Importantly, this behavior is an intended safety feature rather than a failure mode: the operator is constructed to avoid persistent bias and to intervene only when the trajectory is likely at risk. Consistent with this design, we observe more consistent benefits in regimes where uncertainty is elevated and drift is more likely, such as long-horizon reasoning chains (Figure 4) and distractor/perturbation robustness evaluations (Table 3 and Figure 5). From a systems perspective, strengthening TiF in low-entropy regimes may require re-calibrating the demand signal (e.g., a learned monotone mapping of entropy, normalization across vocabularies/temperatures, or alternative uncertainty estimates) so that the gating remains sensitive without sacrificing do-no-harm behavior. Finally, TiF introduces per-token overhead due to the controller update and (optionally) ODE integration. In our default configuration, we use a compute-lean fixed-step Euler update ($N = 1$) implemented inline in the decoding loop. For higher-precision solver ablations (e.g., RK4/RK45 with adjoint sensitivity), we rely on `torchdiffeq`. However, general-purpose ODE solvers are not optimized for the small, high-frequency per-token integration pattern in LLM decoding, and their overhead can be non-negligible. Developing more specialized, hardware-efficient integrators (e.g., fused kernels, closed-form/approximate updates for dissipative dynamics, or compiler-optimized solvers) is an important direction for future work.

### B.2. Scale Effects: Llama-3.1-8B GSM8K Step Buckets and Flip Outcomes

To complement the 3B analysis in the main text, we report depth-stratified GSM8K results for the larger Llama-3.1-8B model and decompose the per-bucket changes into paired outcome flips. Overall, the 8B model shows the same qualitative

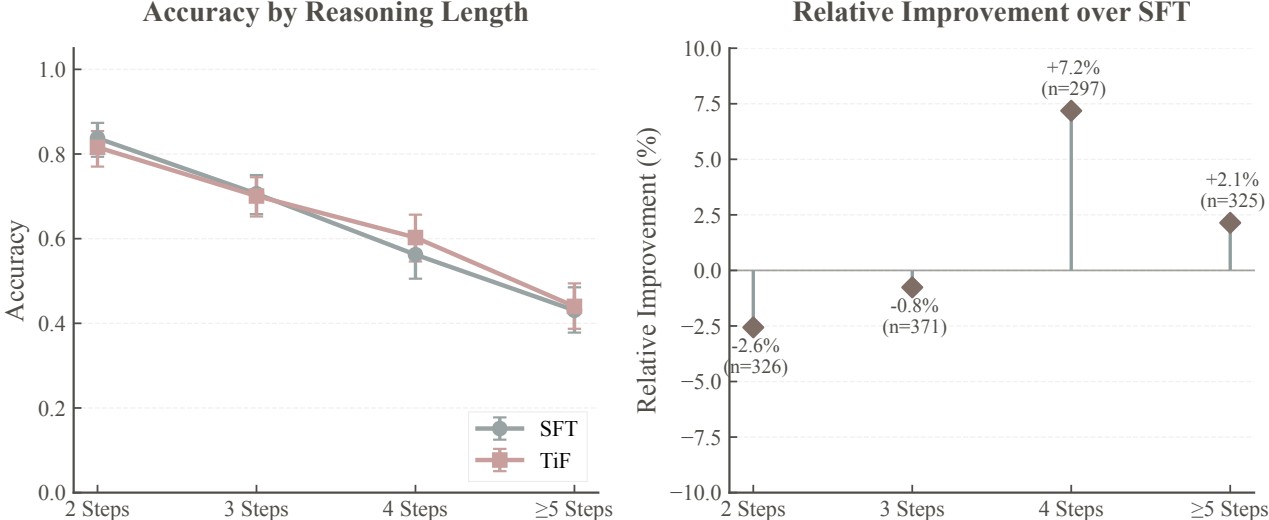

*Figure 7.* **GSM8K step-bucket accuracy for Llama-3.1-8B.** For each reasoning-length bucket, we report accuracy of the SFT baseline and TiF, along with the relative improvement annotated above the TiF curve (e.g., small negative deltas on 2-step problems and positive deltas on 4-step and $\geq$5-step problems, with bucket sizes $n$ in parentheses). Short 2–3 step buckets show small fluctuations around the baseline, while modest gains appear on longer chains, consistent with TiF acting primarily as a stabilization operator for harder instances.

pattern as the 3B model—gains concentrate on harder, longer-horizon instances—but with smaller average effect sizes, consistent with the stronger base model leaving less headroom.

**Step-bucket accuracy (8B).** Figure 7 shows SFT vs. TiF accuracy on GSM8K for Llama-3.1-8B, stratified by reasoning length. Short 2–3 step buckets exhibit small fluctuations around the SFT baseline (within a few percentage points), and in some cases TiF slightly regresses, reflecting its do-no-harm design on already easy instances. In contrast, the 4-step and $\geq$5-step buckets show modest but consistent positive deltas in favor of TiF. This mirrors the 3B behavior and supports the view that the controller mainly helps stabilize longer reasoning chains where error accumulation and drift are more likely.

**Paired flip outcomes (8B).** Figure 8 decomposes the accuracy differences into beneficial and harmful flips for the same GSM8K examples in each bucket. In the 2–3 step buckets, the rates of wrong→right and right→wrong flips are comparable, so the net effect on accuracy is close to zero. For 4-step and $\geq$5-step problems, the fraction of wrong→right flips clearly exceeds the fraction of right→wrong flips, indicating that TiF tends to rescue a subset of difficult instances while only rarely breaking already correct solutions. This flip pattern is consistent with the intended role of TiF as a risk-triggered stabilization operator: it intervenes mainly when the trajectory is fragile (long, high-uncertainty chains), and remains close to identity on easier, low-risk cases.

### B.3. Empirical Stability Diagnostics

This section provides empirical diagnostics that connect the learned controller to the sufficient stability conditions used in Section 4. While our theorems rely on global (and potentially conservative) bounds, the plots below probe the *local* behavior on states visited during inference, i.e., the region of practical interest (the "reasoning manifold").

**Local Lipschitzness of the learned drive $g_\theta$ (supports $\gamma > L_s$).** Figure 9 reports the empirical distribution of the Jacobian spectral norm $\|\partial g_\theta / \partial s\|_2$ on sampled inference states. Interpreting $\|\partial g_\theta / \partial s\|_2$ as a local proxy for the state-Lipschitz constant, the histogram concentrates well below the damping level $\gamma$ (shown as a vertical reference line). This indicates that, on typical inference trajectories, the driven term is locally non-expansive relative to the dissipative component $-\gamma s$, leaving a practical margin for contraction. This diagnostic therefore complements the sufficient condition $\gamma > L_s$ used in the dissipativity and incremental-stability analyses.

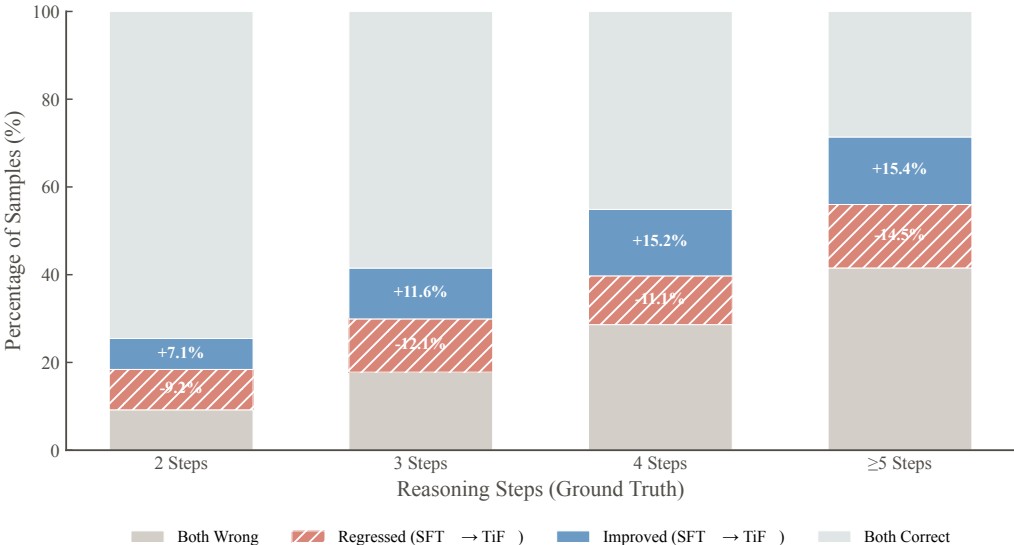

*Figure 8.* **Flip outcomes for Llama-3.1-8B on GSM8K by reasoning length (greedy@1).** Each grouped block shows, within a given step bucket, the percentage of examples that are both correct, improved (SFT wrong → TiF correct), regressed (SFT correct → TiF wrong), or both wrong. In shorter buckets, improved and regressed cases nearly balance, whereas for 4-step and ≥5-step problems the improved flips dominate, leading to a small but consistent net gain on the hardest instances.

**Bounded feature interface $\|\varphi_t\|_2$ (supports bounded-input assumptions).** Our stability statements treat the context feature $\varphi_t$ as a bounded exogenous input. Figure 10 shows that $\|\varphi_t\|_2$ remains tightly controlled across reasoning steps, with rare outliers still bounded (e.g., p99 $\approx 27.07$, p99.9 $\approx 27.76$, and max $\approx 27.97$ in this diagnostic run). This empirical boundedness supports the modeling assumption that the controller is driven by inputs in a compact set, which is a prerequisite for uniform boundedness arguments (e.g., existence of an absorbing set) and for keeping the controller response calibrated over long contexts.

**Contraction of the implemented discrete flow map and its robustness to feature magnitude.** Beyond component-wise diagnostics, we directly probe the *implemented* one-step update map (the discrete flow) by measuring a contraction coefficient for the state transition. Figure 11 correlates $\|\varphi_t\|_2$ with the Jacobian spectral norm of the discrete update map (a contraction metric), where values below 1 indicate local contraction. All sampled points lie comfortably below the contraction threshold, and, crucially, even the high-norm feature outliers ($\|\varphi_t\|_2 > $ p99) exhibit contraction comparable to typical points. This suggests that the realized controller dynamics remain contractive on the inference-state distribution and that local stability is not driven by a small subset of "easy" low-norm inputs. Together, these diagnostics provide practical evidence that the learned dynamics satisfy the intended dissipative/contractive behavior in the regime that matters for long-horizon decoding.

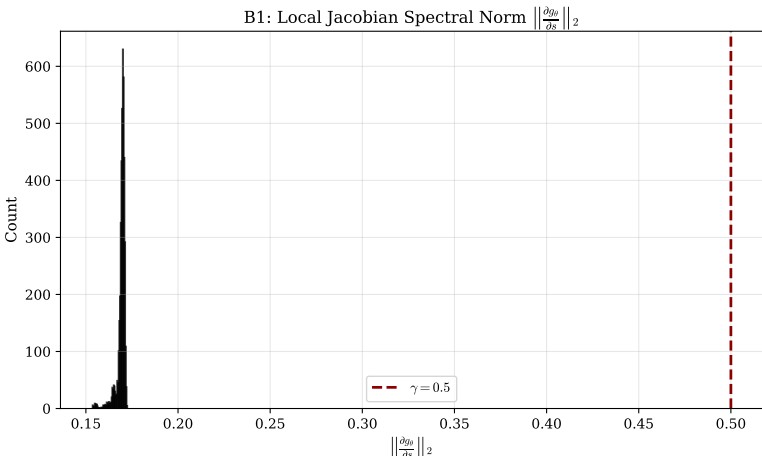

*Figure 9.* Distribution of the Jacobian spectral norm $\|\partial g_\theta / \partial s\|_2$. This diagnostic checks whether the learned drive $g_\theta$ satisfies the Lipschitz condition $\gamma > L_s$ locally on the data distribution.

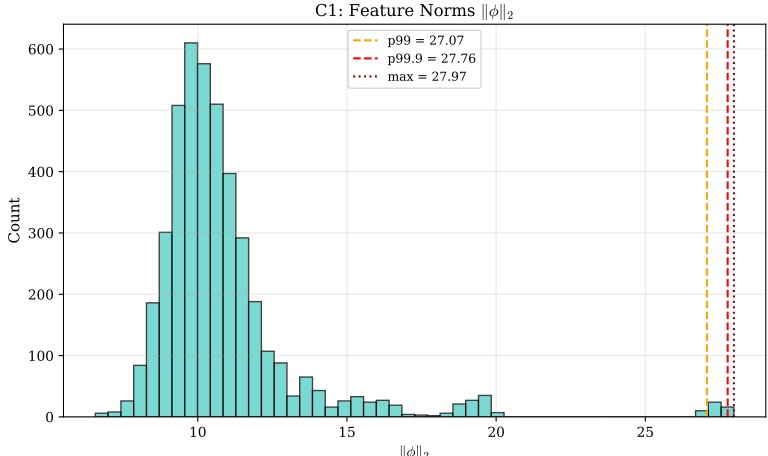

*Figure 10.* Distribution of input feature norms $\|\varphi_t\|_2$ across reasoning steps. Bounded features are a prerequisite for the uniform boundedness guarantees.

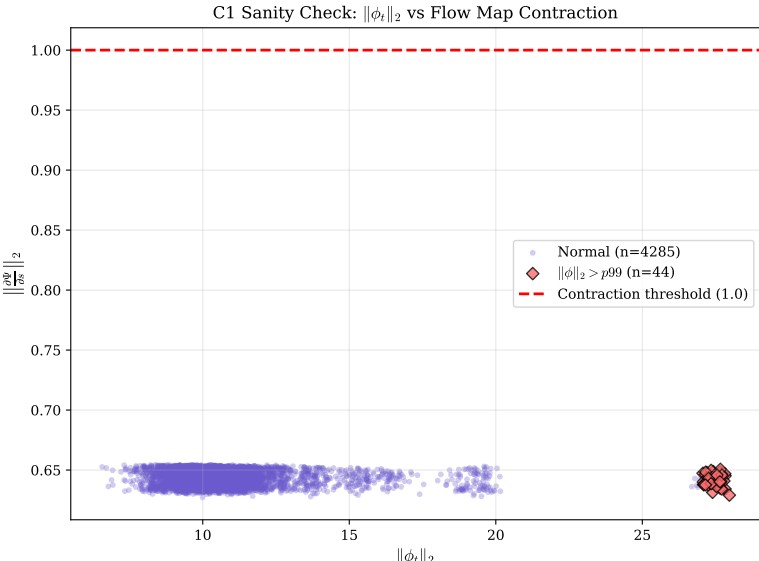

*Figure 11.* Scatter plot correlating feature norms with the contraction coefficient. This provides a fine-grained view of how input magnitude interacts with stability.

*Table 6.* Definitions of perturbation categories in GSM-Plus (Li et al., 2024) for robustness evaluation.

| Perturbation | Description |
|---|---|
| Numerical Substitution | It involves changing some numerical data while minimizing alterations to the textual aspects, ensuring the question's validity remains intact. |
| Digit Expansion | It refers to the process of increasing the number of digits of some numerical values while minimizing alterations to the textual aspects, ensuring the question's validity remains intact. |
| Integer-decimal-fraction Conversion | It refers to the process of converting some integer numbers into decimal or fractional representations while trying to keep the textual aspects unchanged, ensuring that the validity of the question is maintained. |
| Adding Operation | It involves adding extra statements to the original problems, thereby increasing the number of reasoning steps or operations required to solve the rewritten question. The allowed operations are limited to addition, subtraction, multiplication, and division. |
| Reversing Operation | It refers to converting the required answer of the original question into a known condition and transforming one known variable into the newly desired answer while avoiding the introduction of additional constraints. As a result, the rewritten question yields a distinct required answer compared to the original solution. |
| Problem Understanding | It refers to transforming the original problem into a new problem that uses different wording or different sentence structures but does not change the solution of the original problem. |
| Distractor Insertion | It involves introducing distracting conditions that have no impact on the final answer. These introduced conditions should be relevant to the topic of the original question and preferably include numerical values. However, the rewritten problem must maintain an identical solution to that of the original problem. |

## C. Additional Experiments, Full Results, and Definitions

### C.1. GSM-Plus Perturbation Category Definitions

In our GSM-Plus evaluation (Section 5), each example is constructed by applying a targeted perturbation to a base GSM8K-style problem. Table 6 summarizes the perturbation categories and their descriptions.

### C.2. Detailed GSM-Plus Robustness Results

We provide the full numerical breakdown of the GSM-Plus robustness evaluation (Llama-3.2-3B) in Table 7. These results correspond to the visualization in Figure 5 of the main text.

*Table 7.* Accuracy (%) on GSM-Plus (Li et al., 2024) across 7 perturbation categories. Base model: Llama-3.2-3B fine-tuned on GSM8K. **Bold** indicates the best performance, and underline indicates the second best.

| Method | Num. Sub. | Digit Exp. | IDF Conv. | Add. Op. | Rev. Op. | Prob. Underst. | Dist. Ins. |
|---|---|---|---|---|---|---|---|
| SFT | 38.82 | 37.30 | 26.00 | 22.29 | 22.67 | 46.10 | 25.63 |
| NEFTune | 39.35 | 37.45 | 23.12 | 20.47 | 23.43 | 45.41 | 27.29 |
| Pause (Prefix) | 42.68 | 38.67 | 25.63 | 21.61 | 23.73 | 48.07 | 23.96 |
| Pause (Dense) | 40.41 | 36.47 | 24.94 | 22.82 | 23.20 | 45.56 | 25.55 |
| MFT | 43.82 | **40.49** | 29.40 | 23.58 | **24.94** | 47.38 | 28.20 |
| RE-Control | 36.69 | 32.90 | 22.52 | 17.06 | 21.23 | 42.08 | 23.35 |
| TiF (ours) | **43.82** | 39.12 | **30.86** | **24.03** | 23.50 | **49.81** | **28.81** |

## C.3. Descriptions of BIG-Bench Hard (BBH) Tasks

A detailed description of the 6-task BBH subset used in our experiments is provided in Table 8.

*Table 8.* Description of the 6-task BBH subset.

| Task Identifier | Description | Samples |
|---|---|---|
| Logical Deduction (3 Objects) | Ordering three objects based on clues. | 250 |
| Logical Deduction (5 Objects) | Ordering five objects based on clues. | 250 |
| Logical Deduction (7 Objects) | Ordering seven objects based on clues. | 250 |
| Reasoning about Colored Objects | Reasoning about the color of objects in a given context. | 250 |
| Tracking Shuffled Objects (3 Objects) | Tracking the position of three objects after shuffling. | 250 |
| Tracking Shuffled Objects (5 Objects) | Tracking the position of five objects after shuffling. | 250 |

## C.4. Impact of ODE Solver Choice

Unless otherwise noted, we use a fixed-step Euler solver with $N = 1$ as the default configuration in all main experiments; ablations below explore alternative solvers.

We investigate the sensitivity of TiF to the underlying numerical integration scheme. The core component of TiF is the dissipative ODE controller, and the precision of its state evolution $s(\tau)$ could hypothetically affect reasoning stability. In Table 9, we present an ablation study comparing three different solver configurations: a low-precision Euler method, a higher-precision fixed-step RK4 solver, and an adaptive RK45 method. Inference costs for these solvers are reported in Table 10.

Notably, for the scalable RK4 and RK45 configurations, we utilize the *adjoint sensitivity method* (Chen et al., 2018) to compute gradients with $O(1)$ memory cost. Instead of backpropagating through the solver operations, this method solves an augmented ODE backward in time. The adjoint state $\mathbf{a}(t) = \partial L / \partial \mathbf{z}(t)$ follows the dynamics:

$$\frac{\mathrm{d}\mathbf{a}(t)}{\mathrm{d}t} = -\mathbf{a}(t)^\top \frac{\partial f(\mathbf{z}(t), t, \theta)}{\partial \mathbf{z}}. \tag{14}$$

All comparisons use the same Llama-3.2-3B base model.

*Table 9.* **Ablation study on ODE numerical solvers.** We evaluate the impact of integration precision and strategy on reasoning performance using Llama-3.2-Base. We compare three configurations: (1) a simple **Euler** method (single step), (2) an **Adaptive RK45** (Dopri5) solver using the adjoint sensitivity method for memory-efficient backpropagation, (3) a fixed-step **RK4** solver ($N = 10$) also utilizing the adjoint method. Results are reported on the **GSM8K** benchmark.

| ODE Solver Scheme | Adjoint | greedy@1 | maj@8 |
|---|---|---|---|
| SFT (Baseline) | ✗ | 44.73 | 53.30 |
| Euler (Fixed, $N = 1$) | ✗ | 50.49 | 58.38 |
| Euler (Fixed, $N = 2$) | ✗ | 49.20 | 57.85 |
| RK4 (Fixed, $N = 10$) | ✓ | 48.14 | 57.24 |
| RK45 (Adaptive, Dopri5) | ✓ | 48.29 | 57.23 |

*Table 10.* Inference cost comparison for different ODE solvers (Llama-3.2-3B on GSM8K). We report Total TFLOPs, FLOPs per Token, and generation throughput (Tokens/sec).

| Solver | Total TFLOPs | FLOPs/Token | Tokens/sec |
|---|---|---|---|
| Euler (Fixed, $N = 1$) | 532.13 | 4.66G+3.93M | 134.54 |
| RK4 (Fixed, $N = 10$) | 2057.05 | 4.66G+11.01M | 99.61 |
| RK45 (Adaptive) | 2300.61 | 4.66G+11.01M | 106.65 |

## C.5. Additional Ablation Studies

### C.5.1. SENSITIVITY ANALYSIS OF DAMPING COEFFICIENT $\gamma$

We analyze the impact of the damping coefficient $\gamma$ on reasoning performance. The coefficient $\gamma$ controls the stability of the latent dynamics, where higher values imply stronger contraction and stability. We evaluate TiF on GSM8K with $\gamma \in \{0, 0.5, 1.5\}$ and present the results in Table 11.

*Table 11.* Ablation study on the damping coefficient $\gamma$ (GSM8K).

| $\gamma$ | greedy@1 | maj@8 |
|---|---|---|
| 0.0 | 3.71 | – |
| 0.5 | 50.49 | 58.38 |
| 1.5 | 49.73 | 57.31 |

# D. Reproducibility, Implementation, and Efficiency

## D.1. Efficiency Analysis

We conduct an efficiency analysis to compare the computational cost of our proposed method (TiF) against the SFT baseline. To ensure a fair comparison, the SFT baseline uses the exact same training configuration (e.g., batch size, learning rate schedule, and hardware environment) as TiF for each corresponding model and dataset. Table 12 presents the total training time and training throughput (samples per second) for both methods across different base models and datasets. Table 13 summarizes the parameter overhead introduced by the ODE controller, which is negligible ($< 0.2\%$) across all models. Furthermore, Table 14 details the inference cost for Llama-3.2-3B on GSM8K, reporting Total TFLOPs, FLOPs/Token, and throughput. Unless otherwise stated, all inference-time efficiency measurements use a batch size of 8.

## D.2. Algorithmic Description of TiF Decoding

We provide a detailed step-by-step description of the TiF decoding process in Algorithm 1. This algorithm outlines the interaction between the base LLM and the dissipative ODE controller, covering the full cycle from base readout and uncertainty estimation to the gated state injection and continuous latent state update.

TiF is a hybrid discrete–continuous process: for each token step $t$, we evolve a continuous thought state $s(\tau)$ over inner-time $\tau \in [0, 1]$ and obtain $s_{t+1} = s(1)$. In all main experiments, we use a compute-lean solver (Euler, fixed-step $N = 1$), while higher-precision flows (RK4, RK45) are evaluated in solver ablations.

## D.3. Training Configuration and Reporting Checkpoints

**Implementation.** All experiments are implemented with LLaMA-Factory (Zheng et al., 2024). Table 15 summarizes the full training configuration and reporting checkpoints.

**Protocol.** We employ a fixed budget of 10 epochs for both datasets, corresponding to 590 optimizer updates on GSM8K and 1180 updates on MATH due to differing effective batch sizes. We follow a two-stage training schedule: (i) freeze the base LLM and warm up the ODE controller for 2 epochs; (ii) jointly fine-tune both components for the remaining epochs. Reporting checkpoints were determined by monitoring training loss convergence: for the 3B and 4B model families, we selected checkpoints where the training loss stabilized around approximately $0.1$; for the larger Llama-3.1-8B, consistent with neural scaling laws, the model achieved a lower loss due to increased capacity, and was thus evaluated at a later

*Table 12.* Efficiency comparison between SFT and TiF. We report the total training time (hours) and training throughput (samples/second). For fair comparison, SFT baselines share the same training configuration as TiF.

| Base model | Train | Method | Total Training Time (hours) | Throughput (samples/s) |
|---|---|---|---|---|
| Qwen3-4B | GSM8K | SFT | 2.35 | 8.84 |
| | | TiF (ours) | 2.38 | 8.72 |
| | MATH | SFT | 4.53 | 4.60 |
| | | TiF (ours) | 4.69 | 4.44 |
| Qwen2.5-3B | GSM8K | SFT | 1.75 | 11.87 |
| | | TiF (ours) | 1.83 | 11.32 |
| | MATH | SFT | 3.38 | 6.15 |
| | | TiF (ours) | 3.64 | 5.64 |
| Llama-3.2-3B | GSM8K | SFT | 1.42 | 14.62 |
| | | TiF (ours) | 1.84 | 11.29 |
| | MATH | SFT | 3.47 | 6.00 |
| | | TiF (ours) | 3.75 | 5.56 |
| Llama-3.1-8B | GSM8K | SFT | 4.41 | 4.71 |
| | | TiF (ours) | 4.14 | 5.02 |
| | MATH | SFT | 8.53 | 2.44 |
| | | TiF (ours) | 8.22 | 2.53 |

*Table 13.* ODE Controller Parameter Overhead. Counts represent the total number of parameters. **Base** refers to the parameters of the base LLM, while **Extra** refers to the parameters of our ODE controller.

| Model | Base Params | Extra (ODE Controller) | Total Params | Overhead |
|---|---|---|---|---|
| Qwen3-4B | 4,412,839,809 ($\sim$4.41B) | 1,745,793 ($\sim$1.75M) | 4,414,585,602 ($\sim$4.41B) | +0.0396% |
| Qwen2.5-3B | 3,398,256,001 ($\sim$3.40B) | 1,416,577 ($\sim$1.42M) | 3,399,672,578 ($\sim$3.40B) | +0.0417% |
| Llama-3.2-3B | 3,608,430,977 ($\sim$3.61B) | 2,075,009 ($\sim$2.08M) | 3,610,505,986 ($\sim$3.61B) | +0.0575% |
| Llama-3.1-8B | 8,041,288,193 ($\sim$8.04B) | 13,128,193 ($\sim$13.1M) | 8,054,416,386 ($\sim$8.05B) | +0.1633% |

checkpoint (Epoch 7) to ensure comparable convergence maturity. Crucially, within each base model, all methods (SFT, TiF, and baselines) share the exact same training configuration and are evaluated at the **identical reporting checkpoint** to ensure a strictly fair comparison.

# E. Detailed Theoretical Proofs

This appendix provides full proofs for the well-posedness, dissipativity, and incremental stability results stated in Section 4. Our arguments rely on standard results from ODE theory and nonlinear control systems (Coddington et al., 1956; Khalil, 2002).

## E.1. Remark on Numerical Discretization

Theorems in Section 4 are stated for the exact flow map $\mathbf{s}_{t+1} = \mathbf{s}(1)$. When a numerical solver is used, the implemented update can be written as $\widehat{\mathbf{s}}_{t+1} = \widehat{\Psi}(\mathbf{s}_t, \boldsymbol{\varphi}_t)$, where $\Psi(\mathbf{s}_t, \boldsymbol{\varphi}_t)$ denotes the exact flow map at $\tau = 1$. For a $p$-th order method with step size $\Delta\tau$ on the fixed interval $[0, 1]$, under standard smoothness/Lipschitz conditions the global map error satisfies $\|\widehat{\Psi}(\mathbf{s}_t, \boldsymbol{\varphi}_t) - \Psi(\mathbf{s}_t, \boldsymbol{\varphi}_t)\|_2 \leq \varepsilon_{\text{solver}}$ with $\varepsilon_{\text{solver}} = K\Delta\tau^p$ for some constant $K$. The stability bounds in Appendix E.3–E.4 then hold with an additional additive $\varepsilon_{\text{solver}}$ term per decoding step, which accumulates geometrically under the same contraction factor $\rho$. Moreover, if the implemented one-step map is contractive with factor $\rho \in (0, 1)$ on the relevant inference-state region and the per-step solver error satisfies $\|\widehat{\Psi} - \Psi\|_2 \leq \varepsilon_{\text{solver}}$, then the accumulated deviation is bounded by a geometric series, yielding an $O(\varepsilon_{\text{solver}}/(1 - \rho))$ steady-state error bound.

*Table 14.* Inference cost comparison for Llama-3.2-3B on GSM8K. We report Total TFLOPs, FLOPs per Token, and generation throughput (Tokens/sec).

| Method | Total TFLOPs | FLOPs/Token | Tokens/sec |
|---|---|---|---|
| SFT | 548.47 | 4.66G | 145.17 |
| TiF (ours) | 532.13 | 4.66G+3.93M | 134.54 |

---

**Algorithm 1** TiF decoding step with a dissipative Neural ODE controller (one token).

---

**Require:** Base Transformer with LM head $\mathbf{W}_{\text{head}}$; controller parameters $(\mathbf{U}_h, \mathbf{W}_p, \mathbf{W}_r, \theta)$; damping $\gamma > 0$; global scale $\alpha_0 \geq 0$; capacity scale $\lambda_{\text{cap}} > 0$; numerical solver ODESOLVE$(\cdot)$.
**Require:** Current token prefix $x_{\leq t-1}$ and controller state $\mathbf{s}_t$.
**Ensure:** Next token $x_t$ and updated controller state $\mathbf{s}_{t+1}$.
1: **Base readout:** compute post-norm readout $\mathbf{h}_t \leftarrow$ TRANSFORMERREADOUT$(x_{\leq t-1})$.
2: **Controller input:** $\boldsymbol{\varphi}_t \leftarrow \mathbf{U}_h \mathbf{h}_t$.
3: **Pre-injection uncertainty (demand):** $\mathbf{p}_t^{\text{base}} \leftarrow \text{softmax}(\mathbf{W}_{\text{head}} \mathbf{h}_t)$.
4:     $H_t \leftarrow -\sum_{i=1}^{|\mathcal{V}|} p_{t,i}^{\text{base}} \log p_{t,i}^{\text{base}}$.
5: **Controller capacity (supply):** $C_t \leftarrow \tanh(\lambda_{\text{cap}} \|\mathbf{s}_t\|_2)$.
6: **Impact gate:** $\alpha_t \leftarrow \alpha_0 \cdot H_t \cdot C_t$.
7: **Direction-gated injection:** $\mathbf{m}_t \leftarrow \mathbf{W}_p \mathbf{s}_t, \quad \mathbf{r}_t \leftarrow \tanh(\mathbf{W}_r[\boldsymbol{\varphi}_t; \mathbf{s}_t]), \quad \tilde{\mathbf{h}}_t \leftarrow \mathbf{h}_t + \alpha_t(\mathbf{r}_t \odot \mathbf{m}_t)$.
8: **Token prediction:** $\mathbf{p}_t \leftarrow \text{softmax}(\mathbf{W}_{\text{head}} \tilde{\mathbf{h}}_t)$; sample or take greedy $x_t \sim \mathbf{p}_t$.
9: **ODE state update:** integrate on inner-time $\tau \in [0,1]$ with fixed $\boldsymbol{\varphi}_t$:
10:     $\dot{\mathbf{s}}(\tau) = -\gamma \mathbf{s}(\tau) + g_\theta(\mathbf{s}(\tau), \boldsymbol{\varphi}_t), \quad \mathbf{s}(0) = \mathbf{s}_t$.
11:     $\mathbf{s}_{t+1} \leftarrow \mathbf{s}(1) = \text{ODESOLVE}(\mathbf{s}_t, \boldsymbol{\varphi}_t; \gamma, \theta)$.
12: **return** $(x_t, \mathbf{s}_{t+1})$.

---

### E.2. Proof of Existence and Uniqueness (Theorem 4.3)

**Theorem 4.3 restated.** *Fix $t$ and $\boldsymbol{\varphi}_t$. Under Assumption 4.1, the IVP (2) admits a unique solution on $\tau \in [0,1]$.*

*Proof.* For fixed $\boldsymbol{\varphi}_t$, define the vector field

$$f(\mathbf{s}) := -\gamma \mathbf{s} + g_\theta(\mathbf{s}, \boldsymbol{\varphi}_t).$$

By Assumption 4.1, the map $\mathbf{s} \mapsto g_\theta(\mathbf{s}, \boldsymbol{\varphi}_t)$ is globally Lipschitz with constant $L_s$. Therefore, for all $\mathbf{s}_1, \mathbf{s}_2$,

$$\|f(\mathbf{s}_1) - f(\mathbf{s}_2)\|_2 \leq \gamma \|\mathbf{s}_1 - \mathbf{s}_2\|_2 + \|g_\theta(\mathbf{s}_1, \boldsymbol{\varphi}_t) - g_\theta(\mathbf{s}_2, \boldsymbol{\varphi}_t)\|_2 \leq (\gamma + L_s)\|\mathbf{s}_1 - \mathbf{s}_2\|_2.$$

Hence $f$ is globally Lipschitz in $\mathbf{s}$. By the Picard–Lindelöf theorem (Coddington et al., 1956), the IVP admits a unique solution on any finite interval, in particular on $\tau \in [0,1]$. $\square$

### E.3. Proof of Dissipativity and Uniform Boundedness (Theorem 4.4)

**Theorem 4.4 restated.** *Assume Assumption 4.1 and let $\gamma > L_s$. Define $m_{\text{diss}} := \gamma - L_s > 0$ and $\rho_{\text{diss}} := e^{-m_{\text{diss}}} \in (0,1)$. Then $\|\mathbf{s}_{t+1}\|_2 \leq \rho_{\text{diss}} \|\mathbf{s}_t\|_2 + (1 - \rho_{\text{diss}}) \frac{b_0}{m_{\text{diss}}}$ and consequently $\sup_{t \geq 0} \|\mathbf{s}_t\|_2 \leq \max\{\|\mathbf{s}_0\|_2, b_0/m_{\text{diss}}\}$.*

*Proof.* Fix a decoding step $t$ and suppress dependence on $\boldsymbol{\varphi}_t$ in notation. Consider the Lyapunov function

$$V(\mathbf{s}) \triangleq \frac{1}{2}\|\mathbf{s}\|_2^2.$$

Along a solution $\mathbf{s}(\tau)$ of (2),

$$\frac{\mathrm{d}}{\mathrm{d}\tau} V(\mathbf{s}(\tau)) = \left\langle \mathbf{s}(\tau), \frac{\mathrm{d}\mathbf{s}(\tau)}{\mathrm{d}\tau} \right\rangle = -\gamma \|\mathbf{s}(\tau)\|_2^2 + \langle \mathbf{s}(\tau), g_\theta(\mathbf{s}(\tau), \boldsymbol{\varphi}_t) \rangle.$$

By Cauchy–Schwarz and Assumption 4.1 (specifically the linear growth bound $\|g_\theta\|_2 \leq L_s \|\mathbf{s}\|_2 + b_0$),

$$\langle \mathbf{s}, g_\theta(\mathbf{s}, \boldsymbol{\varphi}_t) \rangle \leq \|\mathbf{s}\|_2 \|g_\theta(\mathbf{s}, \boldsymbol{\varphi}_t)\|_2 \leq \|\mathbf{s}\|_2 (L_s \|\mathbf{s}\|_2 + b_0) = L_s \|\mathbf{s}\|_2^2 + b_0 \|\mathbf{s}\|_2.$$

*Table 15.* Training configuration and reporting checkpoints.

| Item | Value |
|---|---|
| Hardware / batch size | $8 \times$ NVIDIA RTX 4090 (48GB). Per-GPU batch size: 16 (GSM8K), 8 (MATH). |
| Training budget | 10 epochs. Optimizer updates: 590 (GSM8K), 1180 (MATH). |
| Two-stage schedule | Freeze base LLM and train ODE controller for 2 epochs, then jointly fine-tune for the remaining epochs. |
| Optimizer | AdamW with LLaMA-Factory default hyperparameters. |
| Learning rates | Base LLM LR: $5 \times 10^{-6}$. ODE controller LR: $100\times$ base LR. |
| LR schedule | Cosine schedule, no warmup. |
| Cutoff len | Max Cutoff len : 512. |
| Decoding | Self-consistency uses 8 samples (maj@8). |
| ODE solver | Fixed-step Euler ($N = 1$). |
| TiF hyperparameters | $\alpha_0 = 100, \gamma = 0.5$. |
| Controller architecture | For 3B/4B base models: $g_\theta$ is a 2-layer MLP, controller state dim $d_s{=}128$, MLP hidden dim 256. For the 8B base model: $g_\theta$ is a 3-layer MLP, $d_s{=}512$, MLP hidden dim 1024. |
| Reporting checkpoint | Llama-3.2-3B: 5; Llama-3.1-8B: 7; Qwen3-4B: 5; Qwen2.5-3B: 3. |

Therefore,

$$\frac{\mathrm{d}}{\mathrm{d}\tau} V(\mathbf{s}(\tau)) \leq -(\gamma - L_s)\|\mathbf{s}(\tau)\|_2^2 + b_0\|\mathbf{s}(\tau)\|_2.$$

Let $r(\tau) \triangleq \|\mathbf{s}(\tau)\|_2$. For $r(\tau) > 0$, we have $\frac{\mathrm{d}}{\mathrm{d}\tau}r(\tau) = \frac{1}{r(\tau)}\frac{\mathrm{d}}{\mathrm{d}\tau}V(\mathbf{s}(\tau))$. With $m_{\mathrm{diss}} = \gamma - L_s$, this yields the comparison inequality

$$\frac{\mathrm{d}}{\mathrm{d}\tau}r(\tau) \leq -m_{\mathrm{diss}}r(\tau) + b_0 \qquad \text{for almost all } \tau \in [0, 1].$$

(At $r(\tau) = 0$, the inequality still holds by continuity.) Solving $\dot{u} = -m_{\mathrm{diss}}u + b_0$ with $u(0) = r(0) = \|\mathbf{s}_t\|_2$ gives

$$u(\tau) = e^{-m_{\mathrm{diss}}\tau}\|\mathbf{s}_t\|_2 + (1 - e^{-m_{\mathrm{diss}}\tau})\frac{b_0}{m_{\mathrm{diss}}}.$$

By comparison, $r(\tau) \leq u(\tau)$ for $\tau \in [0, 1]$. Evaluating at $\tau = 1$ yields

$$\|\mathbf{s}_{t+1}\|_2 = r(1) \leq e^{-m_{\mathrm{diss}}}\|\mathbf{s}_t\|_2 + (1 - e^{-m_{\mathrm{diss}}})\frac{b_0}{m_{\mathrm{diss}}}.$$

Setting $\rho_{\mathrm{diss}} = e^{-m_{\mathrm{diss}}}$ gives (9). Iterating implies

$$\|\mathbf{s}_t\|_2 \leq \rho_{\mathrm{diss}}^t\|\mathbf{s}_0\|_2 + (1 - \rho_{\mathrm{diss}}^t)\frac{b_0}{m_{\mathrm{diss}}} \leq \max\left\{\|\mathbf{s}_0\|_2, \frac{b_0}{m_{\mathrm{diss}}}\right\},$$

which matches (10). $\qquad\square$

### E.4. Proof of Incremental Stability (Theorem 4.6)

**Theorem 4.6 restated.** *Assume $\gamma > L_s$ and Assumption 4.5. Then the controller exhibits incremental stability as in* (12) *and* (15).

*Proof.* Fix $t$ and consider two feature vectors $\boldsymbol{\varphi}_t, \boldsymbol{\varphi}'_t$ and two corresponding solutions $\mathbf{s}(\tau), \mathbf{s}'(\tau)$ to (2) with initial conditions $\mathbf{s}(0) = \mathbf{s}_t$ and $\mathbf{s}'(0) = \mathbf{s}'_t$. Define $\boldsymbol{\delta}(\tau) \triangleq \mathbf{s}(\tau) - \mathbf{s}'(\tau)$. Subtracting the two dynamics yields

$$\frac{\mathrm{d}\boldsymbol{\delta}(\tau)}{\mathrm{d}\tau} = -\gamma\boldsymbol{\delta}(\tau) + \Big(g_\theta(\mathbf{s}(\tau), \boldsymbol{\varphi}_t) - g_\theta(\mathbf{s}'(\tau), \boldsymbol{\varphi}'_t)\Big).$$

Add and subtract $g_\theta(\mathbf{s}'(\tau), \boldsymbol{\varphi}_t)$:

$$g_\theta(\mathbf{s}(\tau), \boldsymbol{\varphi}_t) - g_\theta(\mathbf{s}'(\tau), \boldsymbol{\varphi}'_t) = \Big(g_\theta(\mathbf{s}(\tau), \boldsymbol{\varphi}_t) - g_\theta(\mathbf{s}'(\tau), \boldsymbol{\varphi}_t)\Big) + \Big(g_\theta(\mathbf{s}'(\tau), \boldsymbol{\varphi}_t) - g_\theta(\mathbf{s}'(\tau), \boldsymbol{\varphi}'_t)\Big).$$

By Assumptions 4.1 and 4.5,

$$\|g_\theta(\mathbf{s}(\tau), \boldsymbol{\varphi}_t) - g_\theta(\mathbf{s}'(\tau), \boldsymbol{\varphi}_t)\|_2 \le L_s \|\boldsymbol{\delta}(\tau)\|_2, \qquad \|g_\theta(\mathbf{s}'(\tau), \boldsymbol{\varphi}_t) - g_\theta(\mathbf{s}'(\tau), \boldsymbol{\varphi}'_t)\|_2 \le L_\varphi \|\boldsymbol{\varphi}_t - \boldsymbol{\varphi}'_t\|_2.$$

Taking norms and using the triangle inequality gives

$$\frac{\mathrm{d}}{\mathrm{d}\tau}\|\boldsymbol{\delta}(\tau)\|_2 \le -(\gamma - L_s)\|\boldsymbol{\delta}(\tau)\|_2 + L_\varphi\|\boldsymbol{\varphi}_t - \boldsymbol{\varphi}'_t\|_2.$$

Let $m_{\mathrm{ctr}} = \gamma - L_s > 0$ and note that $\|\boldsymbol{\varphi}_t - \boldsymbol{\varphi}'_t\|_2$ is constant over $\tau$ at fixed step $t$. Solving the comparison ODE $\dot{u} = -m_{\mathrm{ctr}}u + L_\varphi\|\boldsymbol{\varphi}_t - \boldsymbol{\varphi}'_t\|$ with $u(0) = \|\boldsymbol{\delta}(0)\| = \|\mathbf{s}_t - \mathbf{s}'_t\|$ yields

$$\|\boldsymbol{\delta}(1)\|_2 \le e^{-m_{\mathrm{ctr}}}\|\mathbf{s}_t - \mathbf{s}'_t\|_2 + (1 - e^{-m_{\mathrm{ctr}}})\frac{L_\varphi}{m_{\mathrm{ctr}}}\|\boldsymbol{\varphi}_t - \boldsymbol{\varphi}'_t\|_2.$$

Since $\boldsymbol{\delta}(1) = \mathbf{s}_{t+1} - \mathbf{s}'_{t+1}$, setting $\rho_{\mathrm{ctr}} = e^{-m_{\mathrm{ctr}}}$ gives (12). Iterating the one-step bound yields (15).

$$\|\mathbf{s}_t - \mathbf{s}'_t\|_2 \le \rho_{\mathrm{ctr}}^t\|\mathbf{s}_0 - \mathbf{s}'_0\|_2 + \frac{1 - \rho_{\mathrm{ctr}}}{m_{\mathrm{ctr}}}L_\varphi\sum_{k=0}^{t-1}\rho_{\mathrm{ctr}}^{t-1-k}\|\boldsymbol{\varphi}_k - \boldsymbol{\varphi}'_k\|_2. \tag{15}$$

$\square$

### E.5. Additional Steering-Stability Bound

**Corollary E.1** (Coherent steering under feature perturbations). *Under the conditions of Theorem 4.6 and bounded gates* $0 \le \alpha_t, \alpha'_t \le \alpha_{\max}$, *the steering signal satisfies*

$$\begin{aligned}\|\alpha_t(\mathbf{r}_t \odot \mathbf{W}_p\mathbf{s}_t) - \alpha'_t(\mathbf{r}'_t \odot \mathbf{W}_p\mathbf{s}'_t)\|_2 &\le \alpha_{\max}\|\mathbf{W}_p\|\,\|\mathbf{s}_t - \mathbf{s}'_t\|_2 \\ &+ \|\mathbf{W}_p\|\,\|\mathbf{s}'_t\|_2\,|\alpha_t - \alpha'_t| \\ &+ \alpha_{\max}\|\mathbf{W}_p\|\,\|\mathbf{s}'_t\|_2\,\|\mathbf{r}_t - \mathbf{r}'_t\|_\infty.\end{aligned} \tag{16}$$

### E.6. Proof of Steering Corollaries

*Proof.* For Corollary 4.7, sub-multiplicativity of operator norms gives

$$\|\tilde{\mathbf{h}}_t - \mathbf{h}_t\|_2 = \|\alpha_t(\mathbf{r}_t \odot \mathbf{W}_p\mathbf{s}_t)\|_2 \le \alpha_{\max}\|\mathbf{W}_p\|\,\|\mathbf{s}_t\|_2,$$

and Theorem 4.4 bounds $\|\mathbf{s}_t\|_2$ uniformly.

For Corollary E.1, add and subtract $\alpha_t(\mathbf{r}_t \odot \mathbf{W}_p\mathbf{s}'_t)$ and $\alpha_t(\mathbf{r}'_t \odot \mathbf{W}_p\mathbf{s}'_t)$:

$$\|\alpha_t(\mathbf{r}_t \odot \mathbf{W}_p\mathbf{s}_t) - \alpha'_t(\mathbf{r}'_t \odot \mathbf{W}_p\mathbf{s}'_t)\|_2 \le \|\alpha_t(\mathbf{r}_t \odot \mathbf{W}_p(\mathbf{s}_t - \mathbf{s}'_t))\|_2 + \|(\alpha_t - \alpha'_t)(\mathbf{r}'_t \odot \mathbf{W}_p\mathbf{s}'_t)\|_2 + \|\alpha_t((\mathbf{r}_t - \mathbf{r}'_t) \odot \mathbf{W}_p\mathbf{s}'_t)\|_2,$$

and apply $0 \le \alpha_t, \alpha'_t \le \alpha_{\max}$ together with Theorem 4.6. $\square$

### E.7. Promoting the Lipschitz Conditions via Spectral Norms

The stability requirements $\gamma > a$ (Theorem 4.4) and $\gamma > L_s$ (Theorem 4.6) can be encouraged by controlling the expansion of $g_\theta$. For an $L$-layer MLP with Lipschitz activations $\sigma_k$, a standard bound gives

$$L_s \le \prod_{k=1}^{L}\|\mathbf{W}_k\|_2\,L_{\sigma_k}, \tag{17}$$

and spectral normalization (Miyato et al., 2018) provides a practical way to keep these quantities in a stable regime during training.

### E.8. From Controller-State Stability to Reasoning-Chain Preservation

The main theory establishes dissipativity, incremental stability, and bounded steering for the controller state. We now provide a local closed-loop result that connects these state-space properties to token-level preservation of a correct reasoning chain.

Recall that the TiF intervention at decoding step $t$ is

$$\mathbf{u}_t := \alpha_t\big(\mathbf{r}_t \odot \mathbf{W}_p \mathbf{s}_t\big), \qquad \tilde{\mathbf{h}}_t := \mathbf{h}_t + \mathbf{u}_t, \qquad \tilde{\mathbf{z}}_t := \mathbf{W}_{\mathrm{head}}\tilde{\mathbf{h}}_t.$$

Let

$$\mathcal{T}^\star = \{(\mathbf{h}_t^\star, \mathbf{s}_t^\star, \tilde{\mathbf{h}}_t^\star, \tilde{\mathbf{z}}_t^\star, y_t^\star)\}_{t=0}^T$$

denote a reference correct reasoning trajectory, where $y_t^\star$ is the desired token at step $t$. Define the local readout and controller-state errors

$$e_t^h := \|\mathbf{h}_t - \mathbf{h}_t^\star\|_2, \qquad e_t^s := \|\mathbf{s}_t - \mathbf{s}_t^\star\|_2, \qquad \mathbf{e}_t := \begin{bmatrix} e_t^h \\ e_t^s \end{bmatrix}.$$

**Assumption E.2** (Local readout sensitivity around a reference reasoning trajectory). There exists a neighborhood $\mathcal{N}(\mathcal{T}^\star)$ of the reference reasoning trajectory and a constant $a_h \geq 0$ such that, for all states in this neighborhood,

$$\|\mathbf{h}_{t+1} - \mathbf{h}_{t+1}^\star\|_2 \leq a_h\|\tilde{\mathbf{h}}_t - \tilde{\mathbf{h}}_t^\star\|_2 + d_t,$$

where $d_t \geq 0$ is an exogenous local disturbance term that absorbs teacher-forcing mismatch, sampling noise, prefix mismatch, irrelevant-context interference, or other local closed-loop perturbations.

*Remark* E.3. Assumption E.2 is local rather than global. It does not require an explicit global dynamical model for the full Transformer decoder; it only requires that, in a neighborhood of a correct reasoning trajectory, the next-step readout is Lipschitz with respect to the current stabilized readout. This is the local regularity condition needed to transfer controller-state stability to the decoding trajectory.

**Assumption E.4** (Local steering Lipschitzness). There exist constants $c_h, c_s \geq 0$ such that, in the same neighborhood,

$$\|\mathbf{u}_t - \mathbf{u}_t^\star\|_2 \leq c_h\|\mathbf{h}_t - \mathbf{h}_t^\star\|_2 + c_s\|\mathbf{s}_t - \mathbf{s}_t^\star\|_2.$$

*Remark* E.5. The TiF steering map has the form $\mathbf{u}_t = \alpha_t(\mathbf{r}_t \odot \mathbf{W}_p\mathbf{s}_t)$, with bounded impact gate $\alpha_t$ and elementwise bounded direction gate $\mathbf{r}_t$. Assumption E.4 is a local difference-form strengthening of bounded steering: small perturbations in the readout/controller pair induce proportionally small perturbations in the injected steering signal.

**Assumption E.6** (Positive token margin and local LM-head Lipschitzness). Define the reference stabilized logits

$$\tilde{\mathbf{z}}_t^\star := \mathbf{W}_{\mathrm{head}}\tilde{\mathbf{h}}_t^\star.$$

For each step $t$, define the reference logit margin

$$m_t := \tilde{\mathbf{z}}_t^\star(y_t^\star) - \max_{y \neq y_t^\star} \tilde{\mathbf{z}}_t^\star(y), \qquad m_{\min} := \min_{0 \leq t \leq T} m_t.$$

Assume $m_{\min} > 0$. In addition, assume that in the same neighborhood,

$$\|\tilde{\mathbf{z}}_t - \tilde{\mathbf{z}}_t^\star\|_\infty \leq L_z\|\tilde{\mathbf{h}}_t - \tilde{\mathbf{h}}_t^\star\|_2$$

for some constant $L_z > 0$.

*Remark* E.7. Assumption E.6 formalizes a standard robustness principle: if the correct token has a strictly positive logit margin along the reference chain, then sufficiently small perturbations of the stabilized readout cannot flip the argmax token. The Lipschitz condition on the LM head is mild, since the final LM head is linear and hence globally Lipschitz.

We first derive a two-dimensional comparison system that couples the readout error and the controller-state error.

**Proposition E.8** (Local closed-loop comparison system). *Under Assumptions E.2–E.4 and the conditions of the incremental-stability theorem, define*

$$m_{\mathrm{ctr}} := \gamma - L_s > 0, \qquad \rho_{\mathrm{ctr}}(\gamma) := e^{-m_{\mathrm{ctr}}} \in (0,1),$$

*and*

$$\beta_\gamma := \frac{1 - \rho_{\mathrm{ctr}}(\gamma)}{m_{\mathrm{ctr}}} L_\phi \|\mathbf{U}_h\|_2.$$

*Then*

$$\mathbf{e}_{t+1} \preceq \mathbf{A}_\gamma \mathbf{e}_t + \begin{bmatrix} d_t \\ 0 \end{bmatrix}, \qquad \mathbf{A}_\gamma := \begin{bmatrix} a_h(1 + c_h) & a_h c_s \\ \beta_\gamma & \rho_{\mathrm{ctr}}(\gamma) \end{bmatrix},$$

*where $\preceq$ denotes elementwise inequality.*

*Proof.* By incremental stability applied to the reference trajectory and the perturbed trajectory,

$$e_{t+1}^s \leq \rho_{\mathrm{ctr}}(\gamma) e_t^s + \frac{1 - \rho_{\mathrm{ctr}}(\gamma)}{m_{\mathrm{ctr}}} L_\phi \|\boldsymbol{\varphi}_t - \boldsymbol{\varphi}_t^\star\|_2.$$

Since $\boldsymbol{\varphi}_t = \mathbf{U}_h \mathbf{h}_t$ and $\boldsymbol{\varphi}_t^\star = \mathbf{U}_h \mathbf{h}_t^\star$,

$$\|\boldsymbol{\varphi}_t - \boldsymbol{\varphi}_t^\star\|_2 \leq \|\mathbf{U}_h\|_2 \|\mathbf{h}_t - \mathbf{h}_t^\star\|_2 = \|\mathbf{U}_h\|_2 e_t^h.$$

Therefore,

$$e_{t+1}^s \leq \rho_{\mathrm{ctr}}(\gamma) e_t^s + \beta_\gamma e_t^h.$$

Next, by Assumption E.2,

$$e_{t+1}^h \leq a_h \|\tilde{\mathbf{h}}_t - \tilde{\mathbf{h}}_t^\star\|_2 + d_t.$$

Since

$$\tilde{\mathbf{h}}_t - \tilde{\mathbf{h}}_t^\star = (\mathbf{h}_t - \mathbf{h}_t^\star) + (\mathbf{u}_t - \mathbf{u}_t^\star),$$

Assumption E.4 gives

$$\|\tilde{\mathbf{h}}_t - \tilde{\mathbf{h}}_t^\star\|_2 \leq (1 + c_h) e_t^h + c_s e_t^s.$$

Thus,

$$e_{t+1}^h \leq a_h(1 + c_h) e_t^h + a_h c_s e_t^s + d_t.$$

Stacking the two inequalities proves the claim. $\qquad \square$

**Theorem E.9** (Local closed-loop contraction)**.** *Assume the hypotheses of Proposition E.8. If*

$$\rho(\mathbf{A}_\gamma) < 1,$$

*then there exist constants $C_A \geq 1$ and $\nu_\gamma \in (0, 1)$ such that for all $t \geq 0$,*

$$\|\mathbf{e}_t\|_2 \leq C_A \nu_\gamma^t \|\mathbf{e}_0\|_2 + \frac{C_A}{1 - \nu_\gamma} \sup_{0 \leq k < t} d_k.$$

*Proof.* From Proposition E.8,

$$\mathbf{e}_{t+1} \preceq \mathbf{A}_\gamma \mathbf{e}_t + \mathbf{b}_t, \qquad \mathbf{b}_t := \begin{bmatrix} d_t \\ 0 \end{bmatrix}.$$

By induction,

$$\mathbf{e}_t \preceq \mathbf{A}_\gamma^t \mathbf{e}_0 + \sum_{k=0}^{t-1} \mathbf{A}_\gamma^{t-1-k} \mathbf{b}_k.$$

Since $\rho(\mathbf{A}_\gamma) < 1$, the matrix $\mathbf{A}_\gamma$ is Schur stable. Hence there exist $C_A \geq 1$ and $\nu_\gamma \in (0, 1)$ such that

$$\|\mathbf{A}_\gamma^n\|_2 \leq C_A \nu_\gamma^n, \qquad \forall n \geq 0.$$

Taking Euclidean norms and using $\|\mathbf{b}_k\|_2 = d_k$ yields

$$\|\mathbf{e}_t\|_2 \leq C_A \nu_\gamma^t \|\mathbf{e}_0\|_2 + C_A \sum_{k=0}^{t-1} \nu_\gamma^{t-1-k} d_k \leq C_A \nu_\gamma^t \|\mathbf{e}_0\|_2 + \frac{C_A}{1 - \nu_\gamma} \sup_{0 \leq k < t} d_k.$$

$\qquad \square$

**Theorem E.10** (Reasoning-chain preservation under margin)**.** *Assume the hypotheses of Theorem E.9 and Assumption E.6. Define*

$$\kappa_{hs} := \sqrt{(1 + c_h)^2 + c_s^2}.$$

*If for all $0 \le t \le T$,*

$$L_z \kappa_{hs} \|\mathbf{e}_t\|_2 < \frac{m_{\min}}{2},$$

*then*

$$\arg\max_y \tilde{\mathbf{z}}_t(y) = y_t^\star, \qquad \forall\, 0 \le t \le T.$$

*Thus the generated token-level reasoning chain coincides with the reference chain $\{y_t^\star\}_{t=0}^T$, and the final answer is preserved.*

*Proof.* By Assumption E.6,

$$\|\tilde{\mathbf{z}}_t - \tilde{\mathbf{z}}_t^\star\|_\infty \le L_z \|\tilde{\mathbf{h}}_t - \tilde{\mathbf{h}}_t^\star\|_2.$$

Moreover,

$$\|\tilde{\mathbf{h}}_t - \tilde{\mathbf{h}}_t^\star\|_2 \le (1 + c_h)e_t^h + c_s e_t^s \le \kappa_{hs} \|\mathbf{e}_t\|_2,$$

where the last step follows from Cauchy–Schwarz. Hence

$$\delta_t := \|\tilde{\mathbf{z}}_t - \tilde{\mathbf{z}}_t^\star\|_\infty \le L_z \kappa_{hs} \|\mathbf{e}_t\|_2 < \frac{m_{\min}}{2} \le \frac{m_t}{2}.$$

Fix any $t$ and any incorrect token $y \ne y_t^\star$. Then

$$\tilde{\mathbf{z}}_t(y_t^\star) \ge \tilde{\mathbf{z}}_t^\star(y_t^\star) - \delta_t, \qquad \tilde{\mathbf{z}}_t(y) \le \tilde{\mathbf{z}}_t^\star(y) + \delta_t.$$

Therefore,

$$\tilde{\mathbf{z}}_t(y_t^\star) - \tilde{\mathbf{z}}_t(y) \ge \tilde{\mathbf{z}}_t^\star(y_t^\star) - \tilde{\mathbf{z}}_t^\star(y) - 2\delta_t \ge m_t - 2\delta_t > 0.$$

Thus $y_t^\star$ remains the unique argmax at step $t$. Since this holds for every $t \in \{0, \dots, T\}$, the generated token sequence matches the reference reasoning chain, and the final answer is preserved. $\square$

**Corollary E.11** (Explicit sufficient condition for chain preservation)**.** *Assume the hypotheses of Theorem E.10, and let*

$$\bar{d} := \sup_{0 \le t < T} d_t.$$

*If*

$$L_z \kappa_{hs} \left( C_A \|\mathbf{e}_0\|_2 + \frac{C_A}{1 - \nu_\gamma} \bar{d} \right) < \frac{m_{\min}}{2},$$

*then the conclusion of Theorem E.10 holds for all $0 \le t \le T$.*

*Proof.* From Theorem E.9,

$$\|\mathbf{e}_t\|_2 \le C_A \nu_\gamma^t \|\mathbf{e}_0\|_2 + \frac{C_A}{1 - \nu_\gamma} \bar{d} \le C_A \|\mathbf{e}_0\|_2 + \frac{C_A}{1 - \nu_\gamma} \bar{d}.$$

Substituting this bound into the margin condition of Theorem E.10 proves the claim. $\square$

**Interpretation.** The result provides a conditional token-level preservation guarantee. The earlier dissipativity and incremental-stability results control the controller-state perturbation, while the local comparison system couples this perturbation to the readout error. If the joint error remains small enough that the stabilized-logit perturbation consumes less than half of the reference token margin, the argmax token cannot change. Therefore, the guarantee should be read as:

$$\text{state stability} \Rightarrow \text{local closed-loop contraction} \Rightarrow \text{margin preservation} \Rightarrow \text{reasoning-chain preservation.}$$

This does not claim unconditional global accuracy improvement; rather, it states that TiF preserves a correct reasoning trajectory under bounded local perturbations and a positive token margin.

