# OpenReview forum: "Thinking in Flow: A Dissipative Stabilization Operator for Robust Autoregressive Reasoning"
_ICML.cc/2026/Conference — ICML 2026 spotlight_

### Official Review · Reviewer_4qSg · 2026-02-24

**Soundness:** 3
**Presentation:** 3
**Significance:** 2
**Originality:** 3
**Overall Recommendation:** 5
**Confidence:** 4

**Summary:**

The paper focus on the issues of distribution shift and stability in CoT reasoning: irrelevant context and accumulated errors in long-chain reasoning can degrade performance. The authors introduce Thinking in Flow (TiF), which corrects the decoding process by equipping the model with a Neural ODE controller and jointly training it with the base LLM. The core idea is to extract features $\phi_t$ from the LLM’s latent representation $h_t$, solve a dissipative dynamics via an ODE solver to control the controller state $s_t$, and thereby obtain a shift for $h_t$. Additionally, to minimally disrupt the normal decoding process, the shift intensity is adaptively determined by considering both the entropy and the accumulated information in $s_t$. Theoretical results demonstrate its robustness to perturbations, and comprehensive experiments confirm the effectiveness.

**Compliance With Llm Reviewing Policy:**

Affirmed.

**Final Justification:**

I believe the authors have addressed all of my concerns. Therefore, I will raise my score to 5.

**Key Questions For Authors:**

1.	Does TiF potentially harm the diversity of the outputs or limit the capability boundary (refer to the reference 1)? Please present whether the entropy of the output decreases and metrics like Pass@K.
2.	If fine-tuning is conducted simultaneously on datasets of varying difficulty, can TiF still maintain strong performance across all difficulty levels? This scenario is usually more realistic in practice.
3.	Can TiF effectively generalize to out-of-distribution data? Could the constraints imposed on CoT potentially inhibit the LLM’s ability to generalize to more difficult data (e.g., AIME)?
4.	Figure 4 illustrates the differences between TiF and SFT across different reasoning steps, but the definition of a "step" is somewhat ambiguous. It is recommended that the authors use token counts as a more intuitive measure for demonstration.
5.	Have the authors considered training only the controller while keeping the base model frozen (e.g., using a well-tuned SFT model)? To what extent will this degrade performance? In practice, a well-tuned SFT model may already be available, and training only the controller could increase practicality.

Reference:

[1] Yue Y, Chen Z, Lu R, et al. Does reinforcement learning really incentivize reasoning capacity in llms beyond the base model? arXiv preprint arXiv:2504.13837, 2025.

**Limitations:**

yes

**Strengths And Weaknesses:**

Strengths:

1.	Innovatively proposes a Neural ODE controller to mitigate context drift and error accumulation in long-chain reasoning. The paper is well-written and easy to follow.
2.	Despite the complexity of the method design, it consistently improves performance without significantly compromising training or inference efficiency.
3.	The authors provide theoretical results demonstrating TiF’s stability against perturbations and support the assumptions with empirical evidence, showing that each assumption is reasonable.
4.	The authors present comprehensive experimental results covering various reasoning tasks (both mathematical and non-mathematical), robustness to perturbations, and thorough ablation studies to validate the contribution of each module.

Weaknesses:

1.	The authors’motivation is to ensure stability in autoregressive decoding, yet this may potentially impair the diversity of model outputs, thereby narrowing the model’s capability boundary.
2.	Experiments are conducted in a setting where training is performed on datasets aligned with the test set, lacking results on joint training across multiple datasets of varying difficulty. This does not fully align with real-world fine-tuning scenarios. Additionally, it remains unclear whether the controller can generalize to problems of different difficulty levels.
3.	The difficulty of the evaluated datasets peaks at MATH500; there is a lack of analysis on whether TiF remains effective on more challenging problems (e.g., AIME or other complex tasks requiring longer reasoning chains).

---

> ### Author Rebuttal · Authors · 2026-03-30
>
> Thank you for acknowledging the innovative Neural ODE controller, the theoretical stability guarantees, and the high efficiency of our method. We sincerely appreciate your time and effort in reviewing our paper and providing valuable comments. We provide explanations to your questions point-by-point in the following.
> ### **${\color{#f26921}\text{W1 and Q1: Diversity, Capability Boundary, and Pass@K}}$**
> TiF does not impair output diversity or limit the model's capability boundary; empirical metrics show it strictly improves both.
> * Output Entropy: We ran a trajectory-level diagnostic under free-running greedy decoding. At each generated step, we computed the Shannon entropy of the full next-token distribution before taking `argmax`. TiF exhibits higher average token-level predictive entropy than the SFT baseline (macro mean: 0.6066 vs. 0.2868; micro mean: 0.5709 vs. 0.2816). This holds early in generation (first 32 tokens: 0.5009 vs. 0.2716; first 64 tokens: 0.5436 vs. 0.2739), and the paired entropy difference is positive for 99.24% of examples.
> * Pass@K: We evaluated Pass@K on Llama-3.2-3B (GSM8K). As shown below, TiF yields a strictly dominating curve over the SFT baseline across all $k$:
>
> | k | full (pass@k) | ode (pass@k) |
> | :--- | :--- | :--- |
> | 1 | 0.439727 | 0.457165 |
> | 2 | 0.543594 | 0.576952 |
> | 3 | 0.609553 | 0.654284 |
> | 4 | 0.652009 | 0.690675 |
> | 5 | 0.682335 | 0.712661 |
> | 6 | 0.708112 | 0.737680 |
> | 7 | 0.725550 | 0.759666 |
> | 8 | 0.739196 | 0.773313 |
> | 9 | 0.755876 | 0.785444 |
> | 10 | 0.768006 | 0.793025 |
> | 11 | 0.783927 | 0.802123 |
> | 12 | 0.796058 | 0.811221 |
> | 13 | 0.806672 | 0.819560 |
> | 14 | 0.814253 | 0.822593 |
> | 15 | 0.821077 | 0.830174 |
> | 16 | 0.828658 | 0.840030 |
>
> ---
>
> ### **${\color{#f26921}\text{W2 and Q2: Generalization and Joint Training}}$**
> Regarding generalization to different domains (W2), our paper demonstrates that TiF trained on non-mathematical data (CSQA variants) successfully transfers to out-of-domain logical reasoning tasks (the BBH test set), indicating cross-domain robustness.
>
> To address your question regarding joint training on datasets of varying difficulty (Q2), we trained Llama-3.2-3B on a shuffled mixture of the GSM8K (easier) and MATH (harder) training sets, and evaluated on both test sets:
>
> | Dataset | SFT (greedy@1 / maj@8) | TiF (greedy@1 / maj@8) |
> | :--- | :--- | :--- |
> | GSM8K | 45.26 / 54.51 | 46.55 / 54.59 |
> | MATH500 | 12.40 / 16.40 | 10.00 / 17.20 |
>
> As shown above, TiF successfully maintains and generally improves performance across both difficulty levels in a joint training scenario. While there is a slight dip in greedy decoding on the harder MATH500 set, TiF ultimately outperforms the SFT baseline on both datasets under majority voting (maj@8). This confirms its robustness in realistic, mixed-difficulty fine-tuning scenarios.
>
> ---
>
> ### **${\color{#f26921}\text{W3 and Q3: Generalization to Harder Out-of-Distribution Data}}$**
> On AIME, GSM8K-trained Llama-3.2-3B and Qwen3-4B score 0% (maj@32) for both SFT and TiF due to base capacity limits and train/test mismatch, not a TiF penalty. Yet on AMC2024 (harder than MATH500 but solvable by 3B/4B), TiF effectively generalizes. Using 32 rollouts, these GSM8K-trained models yield:
>
> | | Llama-3.2-3B (Trained on GSM8K) | Qwen3-4B (Trained on GSM8K) |
> |:---|:---|:---|
> | SFT | 3.05 (pass@1), 35.0 (pass@32), 7.5 (maj@32) | 20.16 (pass@1), 75.0 (pass@32), 32.5 (maj@32) |
> | TiF | 3.91 (pass@1), 52.5 (pass@32), 10.0 (maj@32) | 20.31 (pass@1), 75.0 (pass@32), 35.0 (maj@32) |
>
> ---
>
> ### **${\color{#f26921}\text{Q4: Definition of "Step" in Figure 4}}$**
> We agree that our current newline-delimited step count is a coarse proxy rather than a fundamental unit of computation. In the revision, we will replace or complement the step-based stratification with token-count-based analysis to provide a more intuitive and reproducible measure of reasoning length.
>
> ---
>
> ### **${\color{#f26921}\text{Q5: Training Only the Controller (Frozen Base Model)}}$**
> We completely agree that freezing a well-tuned SFT model and training only the controller is a highly practical use case. We ran this exact warm-start experiment using Llama-3.2-3B-Instruct on the MATH training set (evaluated on MATH500 greedy).
> * Base Checkpoint: 51.2
> * Frozen Base + Trained TiF Controller: 50.8 (after 1 epoch) / 50.2 (after 2 epochs)
> * Full Fine-Tuning (SFT): drops sharply to 32.0 (after 1 epoch) / 29.8 (after 2 epochs)
>
> This shows that controller-only training preserves most of the performance of an already strong instruct checkpoint (only a 0.4–1.0 point drop), whereas standard full fine-tuning sharply degrades performance due to catastrophic forgetting. We will highlight this in the revision to emphasize TiF's practicality as a lightweight adaptation module.
>
> ---
>
> We appreciate your thoughtful comments. We hope our response addresses your concerns. Please let us know if there are any additional questions, and we will be happy to discuss further.

---

> > ### Author Rebuttal · Reviewer_4qSg · 2026-04-03
> >
> > I believe the authors have addressed all of my concerns. Therefore, I will raise my score to 5.

---

> > > ### Author Response · Authors · 2026-04-03
> > >
> > > Thank you very much for your thoughtful follow-up and for raising your score to 5. We sincerely appreciate your positive assessment and your acknowledgement that our rebuttal has addressed all of your concerns. We are grateful for your support.

---

### Official Review · Reviewer_46LB · 2026-03-10

**Soundness:** 2
**Presentation:** 3
**Significance:** 2
**Originality:** 2
**Overall Recommendation:** 4
**Confidence:** 3

**Summary:**

To mitigate the distraction and inference drift caused by the irrelevant cues during long horizon Chain-of-Thought prompting, this paper proposes Thinking in Flow(TiF): augmenting the standard Transformer with a persistent, continuous-time thought state managed by a lightweight Neural ODE controller. This thought state features dissipative dynamics, allowing the model to accumulate task-relevant evidence while enforcing the controlled forgetting of stale information and perturbations.

The intervention is applied at every single token step to adjust the predicted logits via a residual connection. The intervention is triggered by a demand-supply impact gate to ensure “do-no-harm” corrections and the direction gate determines how to steer in the representation space.

**Compliance With Llm Reviewing Policy:**

Affirmed.

**Final Justification:**

The additional experiments—specifically the non-recurrent ablation and the empirical analysis of the confidence modification—were highly valuable and helped me re-evaluate the proposed method more accurately. Because the intervention occurs at the final stage of next-token prediction (just before the LM head projection), its impact can initially appear somewhat limited, acting more as a local error-correcting module rather than fundamentally altering the global reasoning trajectory. However, considering that the added parameters are highly lightweight and the training overhead is minimal, this efficiency provides practical value that merits publication. Therefore, I have updated my assessment to a slightly positive view.

**Key Questions For Authors:**

1. The introduction of a sequential, continuous-time thought state ($s_t \rightarrow s_{t+1}$) inherently introduces recurrence into the standard Transformer architecture. Could the authors clarify whether the training process for the ODE controller remains parallelizable across the sequence length, or does it require sequential training? A brief discussion on how this recurrence impacts training scalability and GPU utilization would be highly appreciated.

2. The descriptions of the baseline models (e.g., MFT, NEFTune, Pause Tokens, RE-Control) are currently too brief. The reviewer recommend that providing more context on the baselines will help readers better appreciate the comparative advantages of TiF.
3. In Section 4.2 line 364, the authors claim that While the improvement is +6.9% on short tasks (2 steps), it surges to +25.3% on the longest tasks (≥5 steps)."  Is this result calculated as a relative ratio (e.g., $(TiF - SFT) / SFT$) rather than an absolute percentage point difference?

Because the baseline SFT accuracy naturally degrades on longer trajectories, the denominator in this calculation becomes significantly smaller, which can artificially inflate the perceived improvement and could be perceived as an overclaim.

**Limitations:**

yes; the authors discuss technical limitations in Appendix B.1

**Strengths And Weaknesses:**

Strengths

1. The authors mathematically establish the well-posedness, dissipativity, and incremental stability of the controlled thought dynamics. This proves that the model's internal perturbations decay exponentially and interventions remain bounded over arbitrarily long contexts.

2. The proposed TiF architecture is lightweight during inference time; it introduces a negligible parameter overhead of less than 0.2%

Weaknesses

1. At least to this reviewer, it is somewhat ambiguous whether ODE controller maintains a true "thought state" or simply functions as an additional, adaptive layer that steers the output distribution token-by-token. To convincingly isolate the benefits of the recurrent dynamics (passing $s_t$ to $s_{t+1}$) from the mere benefit of adding more tunable parameters, it would be highly valuable to include a non-recurrent parameter-matched ablation. For instance, training a lightweight projection module and establishing residual connection before the LM head projection with the exact same parameter budget would directly answer whether the performance gains are derived from the dynamical stabilization or simply from the standard fine-tuning effect of increased capacity.

2. The proposed dissipative dynamics share strong conceptual similarities with recent advances in linear RNNs and state-space models. In particular, although the authors mention that recurrent or state-space models “do not explicitly design how accumulated evidence should forget stale content or remain contractive under perturbations.” in Related Work in Appendix, it would be beneficial to explicitly discuss and compare the proposed approach with models utilizing forget gates, such as the Gated Delta Net*. How does the continuous-time dissipative gating in TiF compare to the gating mechanisms found in these linear sequence models?
    * Yang, Songlin et al. “Gated Delta Networks: Improving Mamba2 with Delta Rule.” *ArXiv* abs/2412.06464 (2024): n. pag.

3. The "Related Work" section is currently placed entirely in the Appendix. Given its critical importance in contextualizing the paper’s contributions, I strongly recommend moving a condensed version of the Related Work into the main body of the paper.

4. The intervention occurs in the post-norm readout space, just before the language modeling head. Given that a well-trained base model often exhibits high top-1 probabilities, it would be highly appreciated if the authors could explain the intuition behind intervening in this late-stage.

5. Does the stabilization term actively flip the top-1 token prediction, or does it primarily adjust the confidence distribution? It would greatly enhance the paper to include an empirical analysis (or even qualitative case studies on examples) showing exactly *how* the predicted tokens change before and after the stabilizer term is applied. Demonstrating the effect on the token level would provide strong empirical proof to the motivation and intuition behind this method.

---

> ### Author Rebuttal · Authors · 2026-03-30
>
> Thank you for acknowledging the rigorous mathematical establishment of stability, the lightweight architecture, and the negligible inference overhead of our method. We appreciate your valuable comments and address your questions point-by-point below.
>
> ### ${\color{#f26921}\text{W1: Parameter-Matched Non-Recurrent Ablation}}$
>
> We agree a parameter-matched non-recurrent ablation is vital to isolate the benefits of recurrent dynamics. We ran exactly the control you suggested: a lightweight, non-recurrent residual projection layer placed before the LM head, matching TiF's exact parameter budget (+2.08M parameters).
>
> GSM8K results (greedy@1 / maj@8):
> - SFT Baseline: 44.73 / 53.30
> - Non-Recurrent Residual: 48.37 / 57.54
> - TiF (Ours): 50.49 / 58.38
>
> This demonstrates that while adding extra capacity provides some benefit, TiF yields a clear, substantial additional gain over the parameter-matched baseline. The improvement is not driven solely by increased capacity; the persistent, dissipative state dynamics actively contribute to reasoning stabilization. We will include this ablation in the revision.
>
> ### ${\color{#f26921}\text{W2 and W3: Related Work Placement and Relation to Linear RNNs}}$
>
> We strongly agree with your recommendation (W3). Given its critical importance, we will move a condensed Related Work section into the main body of the revision to better contextualize our contributions.
>
> Within this relocated section, we will explicitly foreground our distinction from forget-gated linear models (W2). To clarify here: models like Gated DeltaNet use gating natively within the backbone to manage general sequence memory mixing. TiF, conversely, is a lightweight, post-training *auxiliary stabilization operator* for autoregressive reasoning. TiF's dissipative gate does not primarily decide what backbone context to retain; instead, it couples base-model uncertainty with controller capacity to regulate *when* and *how strongly* to apply a bounded correction to the decoding trajectory.
>
> ### ${\color{#f26921}\text{W4, Q1 and Q2: Post-Norm Intuition, Scalability, and Baselines}}$
>
> Post-Norm Intuition (W4): We intervene here because: (1) It provides a normalized, scale-stable interface directly aligned with the LM head, keeping intervention magnitudes comparable across contexts. (2) It allows us to compute the base model's predictive entropy and apply the correction immediately, without re-running downstream Transformer blocks or altering KV-cache semantics.
>
> Scalability (Q1): While the controller update ($s_t \rightarrow s_{t+1}$) is inherently sequential across tokens, this dependency is strictly confined to a very lightweight auxiliary module. The massive Transformer backbone remains fully parallel. Thus, the added recurrence is highly localized and does not bottleneck GPU utilization.
>
> Baselines (Q2): We will expand the baseline descriptions, grouping them into (1) training-time regularization, (2) computation injection, and (3) test-time representation control, to better highlight TiF's comparative advantages.
>
> ### ${\color{#f26921}\text{W5: Token-Level Effects of Stabilization}}$
>
> To see if TiF primarily adjusts confidence or flips tokens, we compared the next-token distribution immediately before and after the stabilizer during greedy decoding.
>
> TiF primarily sharpens local confidence: the instantaneous intervention decreases entropy on 71.08% of steps and increases the top-1 logit margin on 64.13% of steps. However, it *does* actively flip the top-1 prediction on 8.99% of total steps. Crucially, these flips are highly targeted: the flip rate is 39.68% on highly uncertain steps (high-entropy) versus only 1.24% on low-entropy steps.
>
> Qualitatively, we observe concrete win cases where localized flips rescue downstream reasoning. For example, in one GSM8K trajectory, TiF flipped a crucial token from "points" to "*", correcting a doomed "4 + .25 = 4.25" path into the correct "4 * 1.25 = 5" computation. We will add this empirical analysis to the paper.
>
> ### ${\color{#f26921}\text{Q3: Relative vs. Absolute Improvements}}$
>
> You are completely correct. The "+6.9%" and "+25.3%" values in Figure 4 denote *relative ratios*, not absolute percentage-point gains. We agree relative ratios can appear inflated when the baseline denominator shrinks on harder trajectories, and will explicitly clarify this to avoid overclaiming.
>
> To provide a denominator-free metric, we examined paired outcome flips within each reasoning-depth bucket. For shorter problems (2-3 steps), positive and negative flips are near-neutral. However, for 4-step and $\ge 5$-step problems, TiF successfully converts significantly more wrong answers to correct than vice versa. This confirms TiF’s net benefit is genuinely concentrated on complex, long-horizon instances.
>
> ---
>
> We appreciate your constructive feedback, which has strengthened our claims and positioning. We welcome further discussion.

---

> > ### Author Rebuttal · Reviewer_46LB · 2026-04-03
> >
> > Thanks for the rebuttal. I am raising the score to 4. It is highly recommended to incorporate all your responses into the revision.

---

> > > ### Author Response · Authors · 2026-04-03
> > >
> > > Thank you very much for your thoughtful follow-up and for raising your score. We sincerely appreciate your acknowledgement that our rebuttal has adequately addressed your concerns. Your comments have been highly valuable in helping us strengthen the paper, and we will carefully incorporate the requested clarifications, additional analyses, and related-work discussion into the revision.

---

### Official Review · Reviewer_ePje · 2026-03-11

**Soundness:** 3
**Presentation:** 3
**Significance:** 2
**Originality:** 3
**Overall Recommendation:** 4
**Confidence:** 2

**Summary:**

The paper proposes a theory-guided Thinking in Flow (TiF), which treats decoding as a perturbed long-horizon dynamical system and aims to make the decoding trajectory smooth and confined under distribution shift or context interference.

**Compliance With Llm Reviewing Policy:**

Affirmed.

**Final Justification:**

I think the author has clarified the position of this work, and I would keep my positive score.

**Key Questions For Authors:**

Q1: I would like to know what is the relation of your work and the forgetting gate proposed in various papers.

Q2: I get the motivation for this work, but now it is still difficult for me to get the position of this work in the line of research. As said above, it's hard for me to judge the significance and the originality.

**Limitations:**

yes

**Strengths And Weaknesses:**

In my opinion, the paper is comprehensive and sounded. While I can't judge the novelty of the paper, the experiments is solid and persuasive. The paper also includes the overhead and training time of the method.

---

> ### Author Rebuttal · Authors · 2026-03-30
>
> Thank you for acknowledging that our approach is comprehensive and sound, and that our empirical experiments are solid and persuasive. We sincerely appreciate your time and effort in reviewing our paper and providing valuable comments. We address your questions point-by-point below.
>
> ### **${\color{#f26921}\text{Q1: Relation to Forgetting Gates}}$**
>
> We agree that, at a high level, TiF shares the intuition of controlling information accumulation with forgetting gates found in RNNs or State-Space Models (SSMs). However, their roles, mechanisms, and end goals differ significantly:
>
> 1. **Purpose and Placement:** Standard forgetting gates are intrinsic to the primary sequence-mixing architecture and are updated at every step to manage context memory. In contrast, TiF acts as a lightweight, plug-and-play **stabilization operator** for an *already-trained* Transformer backbone, applying bounded corrections via post-norm residual updates.
> 2. **Selective Intervention:** TiF couples controlled forgetting with strict selective intervention. It steers the hidden representation *only* when the base model is uncertain (entropy-gated) and the controller has accumulated a strong signal. This ensures a strict "do-no-harm" policy, unlike standard recurrent updates.
> 3. **Theoretical Guarantees:** TiF's dissipative formulation is explicitly designed to establish formal control-theory guarantees (well-posedness, dissipativity, and incremental stability) against autoregressive error accumulation.
>
> As another reviewer (Q5H8) accurately noted, TiF integrates these elements into a "unified mechanism rather than a collection of loosely connected tricks." While TiF shares a surface-level similarity with traditional forgetting mechanisms, its contribution is fundamentally a principled, stability-oriented control framework for reasoning.
>
> ---
>
> ### **${\color{#f26921}\text{Q2: Position in the Research Landscape}}$**
>
> We appreciate the opportunity to clarify our positioning. The landscape of LLM reasoning enhancements is indeed crowded, but TiF occupies a distinct intersection of three established research lines:
>
> 1. **Versus Prompting & Decoding Strategies (e.g., ToT, Verification):** Most methods address long-horizon reasoning drift by searching through discrete text paths or injecting pause tokens. TiF fundamentally reframes inference drift as a *dynamical instability* problem. We intervene directly in the continuous latent space to mathematically bound error accumulation before it manifests as fluent but incorrect text.
>
> 2. **Versus Architectural Replacements (e.g., SSMs, Linear RNNs):** While newer architectures introduce controlled memory to handle long contexts natively, they require massive pre-training from scratch. TiF's significance lies in being a lightweight, *post-training plug-in*. It brings continuous-time stable memory to standard, pre-trained discrete Transformers without altering their core architecture.
>
> 3. **Versus Latent Reasoning (e.g., Quiet-STaR):** Recent works explore continuous latent reasoning but often rely on unstructured latent steps. TiF's originality stems from explicitly imposing *dissipative control theory* on this process. By using a Neural ODE with a designed dissipative term, we do not just let the model "think" in the latent space; we guarantee that stale noise decays and interventions remain strictly bounded.
>
> In summary, TiF uniquely bridges discrete text generation and continuous-time control systems as a theoretically grounded, post-training stabilization operator. We hope this breakdown clearly highlights the originality and significance of our contribution.
>
> ---
>
> We appreciate your thoughtful feedback and hope this clarifies TiF's exact positioning and novelty. We welcome any further discussion.

---

> > ### Author Rebuttal · Reviewer_ePje · 2026-04-02
> >
> > The author has resolved my issue and I will keep my positive score.

---

> > > ### Author Response · Authors · 2026-04-03
> > >
> > > Thank you for your follow-up and positive assessment. We are glad that our rebuttal helped clarify the relation to forgetting mechanisms and the positioning of TiF. We sincerely appreciate your acknowledgement that the main concerns have been resolved and your continued positive evaluation.

---

### Official Review · Reviewer_Q5H8 · 2026-03-12

**Soundness:** 3
**Presentation:** 4
**Significance:** 3
**Originality:** 3
**Overall Recommendation:** 4
**Confidence:** 3

**Summary:**

This paper proposes **Thinking in Flow (TiF)**, a lightweight stabilization module for autoregressive reasoning in large language models. The method augments a base Transformer with a low-dimensional persistent thought state whose evolution is modeled by a dissipative Neural ODE, with dynamics of the form `ds/dtau = -gamma s + g_theta(s, phi_t)`. The main intuition is that the dissipative term supports controlled forgetting of stale or noisy reasoning signals, while the learned drive term accumulates task-relevant information over decoding steps.

To keep the intervention selective, TiF injects the controller output back into the model through a post-norm residual update, modulated by two gates: an uncertainty-based impact gate that becomes larger when the base model is uncertain, and a direction gate that constrains how the hidden representation is steered. The authors position this as a "do-no-harm" stabilization operator that should only intervene when needed and remain bounded over long rollouts.

The paper provides both theory and experiments. On the theory side, it analyzes the controller dynamics under Lipschitz-style assumptions and establishes well-posedness, dissipativity, and incremental stability of the controller state, together with bounded intervention magnitude. On the empirical side, it evaluates TiF on mathematical reasoning benchmarks (GSM8K and MATH500) across Qwen and Llama models, compares against several baselines, and studies robustness under irrelevant-context and semantic perturbation benchmarks, as well as a small BBH subset. The reported results show mostly consistent gains, with the strongest improvements on smaller models, longer reasoning chains, and robustness settings.

**Compliance With Llm Reviewing Policy:**

Affirmed.

**Key Questions For Authors:**

Q1. How robust is the method across stronger base models and broader domains?

 The gains are clearly positive on some settings, especially Llama-3.2-3B and the robustness benchmarks, but they become quite small on Llama-3.1-8B and some MATH500 settings. Could the authors clarify their current understanding of when TiF helps most, and whether they have evidence that the stabilization effect persists at larger scales or on more diverse reasoning domains?

 Q2. How often does entropy-based gating fail on confidently incorrect predictions?

 Since TiF uses base-model entropy as the primary demand signal, I would like to better understand what happens when the base model is wrong but confident. Do the authors have empirical observations on how frequently this occurs, and whether learned or richer uncertainty signals materially improve intervention quality?

 Q3. How sensitive are the results to the dissipative strength and controller-state dimensionality?

 The paper includes ablations, but it would be helpful to understand the practical tuning landscape more clearly. Is there a stable operating region for `gamma` and the controller size, or does performance depend on fairly careful hyperparameter selection?

 Q4. To what extent do the controller-state stability guarantees correlate with actual end-task gains?

 The paper provides theoretical and diagnostic evidence for contraction-like behavior. Have the authors examined whether examples or checkpoints with stronger contraction diagnostics also systematically yield better reasoning accuracy or robustness? Such a correlation would strengthen the connection between the theory and the empirical claims.

**Limitations:**

yes

**Strengths And Weaknesses:**

Strengths:

 S1 (Originality / significance). The paper brings a genuinely interesting dynamical-systems perspective to test-time reasoning control. Modeling reasoning regulation as a dissipative stabilization problem is more distinctive than the usual prompting- or token-level intervention framing, and the resulting design is conceptually coherent.

 S2 (Soundness). The method is relatively well motivated and internally consistent. The controller state, post-norm injection, uncertainty-capacity gating, and bounded direction gate fit together as a unified mechanism rather than a collection of loosely connected tricks.

 S3 (Soundness / presentation). The theoretical section is useful and more careful than average for papers of this type. In particular, the paper is explicit that the guarantees concern the controller dynamics and bounded steering, not global closed-loop token-level decoding. That boundary-setting improves credibility.

 S4 (Soundness). The empirical section is reasonably broad. The paper evaluates across multiple base models and both accuracy and robustness settings, includes baseline comparisons, analyzes performance by reasoning depth, and provides ablations for key architectural choices.

 S5 (Presentation). The paper is clearly written and easy to follow. The main intuition, model components, and experimental findings are communicated well, and the figures are helpful for understanding the stabilization story.

 Weaknesses:

 W1 (Soundness). The theoretical guarantees are only indirectly connected to the full claim of improved reasoning. The analysis establishes stability properties of the controller state and bounded intervention under assumptions such as `gamma  L_s`, but it does not show that these properties imply better end-task reasoning in the full closed-loop decoding process. The empirical contraction diagnostics help, but they remain suggestive rather than decisive.

 W2 (Soundness / significance). The empirical gains, while generally positive, are uneven. Improvements are strong for Llama-3.2-3B and some robustness settings, but they are small on larger models such as Llama-3.1-8B and modest on parts of MATH500. This weakens the claim that the method broadly solves long-horizon reasoning brittleness rather than helping in a narrower regime.

 W3 (Soundness). The robustness story is promising but concentrated on a relatively limited set of settings, especially the Llama-3.2-3B GSM8K pipeline. It would strengthen the paper to show whether the same robustness advantages transfer more consistently across additional model families and training setups, rather than mostly on one representative model.

 W4 (Originality / positioning). Although the dynamical framing is novel, some implementation choices are closer to familiar representation-steering or auxiliary-controller methods than the paper's framing might initially suggest. The paper could do a better job of clarifying exactly which aspects are fundamentally new versus new combinations of existing ideas.

 W5 (Presentation / methodology). The uncertainty-based gate is computationally attractive, but entropy is a fairly blunt proxy for intervention need. The paper acknowledges this partly, but it would benefit from a sharper discussion of failure cases, especially confidently wrong predictions where entropy-based triggering may be weak.

 W6 (Significance). The paper makes a strong case that stabilization-style control is promising, but the current evidence does not yet establish TiF as a broadly validated recipe for reasoning improvement across scales and domains. I view the contribution more as a compelling direction with solid initial evidence than as a definitive solution.

---

> ### Author Rebuttal · Authors · 2026-03-30
>
> Thank you for acknowledging the novel dynamical-systems perspective, the conceptual coherence, and the careful theoretical grounding of our method. We sincerely appreciate your time and effort in reviewing our paper and providing valuable comments. We provide explanations to your questions point-by-point in the following.
>
> ---
>
> ### ${\color{#f26921}\text{W1: Bridging Theory to End-Task Reasoning (The New Theorem)}}$
>
> To bridge the gap between state stability and closed-loop decoding, we introduce a new Local Closed-Loop Theoretical Chain.
>
> Assuming a reference trajectory, Lipschitz readout/perturbations, and a positive logit margin, we prove a joint contraction bound for readout and state errors. Crucially, if the bounded perturbation avoids consuming the entire logit margin, the `argmax` of the correct token mathematically cannot flip.
>
> This upgrades our claim to a conditional token-level preservation guarantee: State Stability $\rightarrow$ Local Closed-Loop Contraction $\rightarrow$ Margin Preservation $\rightarrow$ Reasoning-Chain Preservation. The revision will include this proof, clarifying that TiF *preserves* correct reasoning under bounded perturbations, rather than claiming unconditional global accuracy gains.
>
> ---
>
> ### ${\color{#f26921}\text{W2, W3 and Q1: Robustness Across Scales, Domains, and Models}}$
>
> TiF stabilizes trajectories where inference drift is the bottleneck. While average gains on the stable Llama-3.1-8B shrink, paired-flip analysis (App B.2) shows TiF specifically rescues the hardest ($\ge 5$ steps) problems. BBH confirms transfer beyond math.
> * Cross-Model Transfer: Qwen2.5-3B on GSM-Plus shows consistent robustness transfer over SFT: Num.Sub (+0.38), Digit Exp (+1.74), IDF Conv (+2.37), Add.Op (+0.38), Prob.Underst (+2.96), Dist.Ins (+3.33), with a slight drop on Rev.Op (-0.83).
>
> ---
>
> ### ${\color{#f26921}\text{W4: Originality and Positioning}}$
>
> We will explicitly demarcate this boundary in the revision:
> * Fundamentally New: (1) Reframing autoregressive decoding as a perturbed continuous-time dynamical system, (2) the dissipative ODE formulation for controlled forgetting, and (3) providing theoretical stability guarantees.
> * Combinations of Existing Ideas: Our implementation choices—specifically representation-steering (residual injection) and auxiliary controllers (entropy gating)—are indeed familiar techniques. We repurpose these standard mechanisms purely as efficient *delivery vehicles* to instantiate our novel continuous-time dynamics within pre-trained Transformers.
>
> ---
>
> ### ${\color{#f26921}\text{W5 and Q2: Entropy Gating and Confidently Wrong Predictions}}$
>
> We agree entropy is a coarse proxy. In 50 sampled trajectories (1,553 incorrect steps) from Llama-3.2-3B on GSM8K, 80.04% of errors occur at low entropy ($H_t < 0.2$). E.g., confidently misreading "15 cups" as "15 chickens" ($H_t=0.018$) makes downstream arithmetic deterministic ($H_t \sim 10^{-6}$), rendering the gate too weak to intervene. We acknowledge the need for richer uncertainty signals.
>
> ---
>
> ### ${\color{#f26921}\text{Q3: Sensitivity to Dissipative Strength and Controller Dimension}}$
>
> * Damping ($\gamma$): Theory requires $\gamma > L_s$. Empirically, $\gamma \in \{0.5, 1.5\}$ perform well (50.49 & 49.73 greedy@1), while $\gamma = 0$ severely degrades accuracy (3.71%). Over-damping ($\gamma = 5$) collapses performance. Thus, $\gamma$ requires a moderate positive value.
> * Dimension ($d_s$): TiF explicitly targets a lightweight regime ($d_s \ll d_h$). Our parameter-matched control isolates dynamical benefits from capacity increases. We will add a controller-size sensitivity discussion to clarify this.
>
> ---
>
> ### ${\color{#f26921}\text{Q4: Correlation Between Contraction Diagnostics and End-Task Gains}}$
>
> To isolate dynamics from checkpoint confounds, we performed an example-level analysis on the trained Llama-3.2-3B TiF model (GSM8K test). For each trajectory, we computed the average discrete flow-map spectral norm ($\| \frac{\partial \Psi}{\partial s} \|_2$, matching Fig 3) via exact batched SVD. Smaller norms imply stronger contraction.
>
> 1. Correct vs. Incorrect: Correct examples have a lower average norm, yielding a clear negative correlation (Spearman = -0.2995).
> | Outcome | Count | Avg Spectral Norm |
> |:---|---:|---:|
> | Correct | 666 | 0.58260 |
> | Incorrect | 653 | 0.58273 |
>
> 2. Accuracy by Tertiles: Sorting 1,319 test examples into tertiles by spectral norm reveals a strictly monotonic link between contraction and accuracy:
> | Stability Bucket (by Spectral Norm) | Avg Norm | Accuracy |
> |:---|---:|---:|
> | Strong Contraction (Top 33%) | 0.58243 | 68.34% |
> | Moderate Contraction (Mid 33%) | 0.58268 | 47.50% |
> | Weak Contraction (Bot 33%) | 0.58289 | 35.68% |
>
> ---
>
> We appreciate your thoughtful comments. We hope our response addresses your concerns. Please let us know if there are any additional questions, and we will be happy to discuss further.

---

> > ### Author Rebuttal · Reviewer_Q5H8 · 2026-04-03
> >
> > The author has resolved my issue and I will keep my positive score.

---

> > > ### Author Response · Authors · 2026-04-03
> > >
> > > Thank you very much for your thoughtful follow-up. We sincerely appreciate your positive assessment and your acknowledgement that our rebuttal has adequately addressed your concerns. We are grateful for your support.

---

### Decision · Program_Chairs · 2026-04-30

**Decision:**

Accept (spotlight)

**Comment:**

The paper proposes a mechanism to prevent models from drifting away during long CoT reasoning. It introduces a "thought state" governed by dissipative dynamics backed by a lightweight neural ODE controller. This simulates both forgetting stale information and accumulating task-relevant evidence. All reviewers felt that:

1. Treating the dynamics as a dynamical system is an exciting idea different from traditional ideas of prompting or memory management, though some reviewers questioned how it differs from forget-gated RNNs like Gated DeltaNet.
2. The proposed method is also backed up with theory.
3. The architecture is lightweight with only 0.2% increase in parameters.
4. Experiments are thorough with multiple model families.

One major weakness is that the increase in performance primarily depends on model size. Smaller models benefit more, but improvements are modest improvements on large models. The reviewers acknowledged that the rebuttal responses clarified their initial issues.